# Cross-tissue analysis of blood and brain epigenome-wide association studies in Alzheimer's disease

Tiago C. Silva [1], Juan I. Young[2,3], Lanyu Zhang[1], Lissette Gomez[3], Michael A. Schmidt [3], Achintya Varma[3], X. Steven Chen[1,4], Eden R. Martin[2,3] & Lily Wang [1,2,3,4] ✉

To better understand DNA methylation in Alzheimer's disease (AD) from both mechanistic and biomarker perspectives, we performed an epigenome-wide meta-analysis of blood DNA methylation in two large independent blood-based studies in AD, the ADNI and AIBL studies, and identified 5 CpGs, mapped to the *SPIDR*, *CDH6* genes, and intergenic regions, that are significantly associated with AD diagnosis. A cross-tissue analysis that combined these blood DNA methylation datasets with four brain methylation datasets prioritized 97 CpGs and 10 genomic regions that are significantly associated with both AD neuropathology and AD diagnosis. An out-of-sample validation using the AddNeuroMed dataset showed the best performing logistic regression model includes age, sex, immune cell type proportions, and methylation risk score based on prioritized CpGs in cross-tissue analysis (AUC = 0.696, 95% CI: 0.616 – 0.770, *P*-value = 2.78 × 10$^{-5}$). Our study offers new insights into epigenetics in AD and provides a valuable resource for future AD biomarker discovery.

Alzheimer's disease (AD) is the most common cause of dementia and affects about 11% of people 65 years and older in the US[1]. With the rising elderly population, AD has become a major public health concern. In addition to genetics, it has been increasingly appreciated that epigenetic marks such as DNA methylation also play an important role in AD[2–5]. Currently, there is a lack of objective, inexpensive, and minimally invasive biomarkers for AD. Methylated DNA is more stable than mRNA and provides an excellent source of biomarkers[6]. A number of recent studies identified DNA methylation differences in blood samples of AD subjects. Fransquet et al. (2018)[7] conducted a systematic review of 36 studies comparing DNA methylation between AD cases and controls in blood samples and found that 67% of these studies reported significant methylation differences between the groups. Most recently, it was shown that DNA methylation changes could be detected in blood at least three years before the onset of dementia symptoms[8]. In particular, blood methylation levels at several candidate loci[9–13], such as the *COASY*, *HOXB6*, and *APP* genes, were significantly different between AD patients and healthy controls.

In this study, we studied DNA methylation differences in AD from both mechanistic and biomarker perspectives. To help improve power, we meta-analyzed two large blood-based AD EWAS measured by the same Illumina Infinium MethylationEPIC BeadChips, and conducted by the Alzheimer's Disease Neuroimaging Initiative (ADNI)[14] and the Australian Imaging, Biomarkers, and Lifestyle (AIBL)[15] consortiums recently. In addition, we also performed a cross-tissue meta-analysis by combining the DNA methylation datasets measured on blood samples with four additional DNA methylation datasets measured on over one thousand brain prefrontal cortex samples, to prioritize significant methylation differences associated with both AD diagnosis and AD neuropathology. To understand the functional roles of the methylation differences, we conducted several integrative analyses of methylations with genetic variants, gene expressions, and transcription

[1]Division of Biostatistics, Department of Public Health Sciences, University of Miami, Miller School of Medicine, Miami, FL 33136, USA. [2]Dr. John T Macdonald Foundation Department of Human Genetics, University of Miami, Miller School of Medicine, Miami, FL 33136, USA. [3]John P. Hussman Institute for Human Genomics, University of Miami Miller School of Medicine, Miami, FL 33136, USA. [4]Sylvester Comprehensive Cancer Center, University of Miami, Miller School of Medicine, Miami, FL 33136, USA. ✉e-mail: lily.wang@miami.edu

factor binding sites. Moreover, we evaluated the feasibility of the DNA methylation differences as potential biomarkers for diagnosing AD in an external blood sample dataset. Our analysis results provide a valuable resource for future mechanistic and biomarker studies in AD.

## Results

### Study datasets

Our meta-analysis of blood samples included a total of 1284 DNA methylation samples (427 AD cases, 857 cognitive normals) from two large independent AD EWAS, the ADNI[14] and the AIBL[16] studies. The ADNI study is a longitudinal study with DNA methylation samples collected at baseline and multiple follow-up visits ranging from 6 months to 60 months[14]. The average and median times between the first and last visits for the subjects were 18 months and 24 months, respectively. Similarly, the AIBL study is also a longitudinal study with DNA methylation samples collected at 18-month follow-up. A total of 797 unique individuals older than 65 years of age were included in this study. The mean ages of the subjects were 77.13 (±6.64) years and 73.37 (±5.79) years, and the percentages of females were 46% and 54% in the ADNI and AIBL studies, respectively (Supplementary Data 1).

### Meta-analysis identified methylation differences significantly associated with AD at individual CpGs and genomic regions in the blood

After adjusting covariate variables age, sex, batch, and immune cell-type proportions and correcting inflation in each dataset (see details in Methods), the inverse-variance fixed-effects meta-analysis model identified 5 CpGs, mapped to the *SPIDR*, *CDH6* genes, and intergenic regions at 5% false discovery rate (FDR) (Table 1, Supplementary Fig. 1). Among the genes that are associated with these CpGs, the *SPIDR* gene encodes the scaffolding protein involved in DNA repair, which regulates the specificity of homologous recombination to achieve a high degree of fidelity[17]. The *CDH6* gene encodes a cadherins protein, which regulates synaptogenesis and synaptic plasticity[18–20]. Recently, *CDH6* levels in plasma were also shown to be associated with AD in carriers of the *APOE* ε4 allele[21]. The most significant probe cg03429569 is located at 3018 bp upstream of the *HOXD10* gene, a member of the HOX family, which was recently shown to be involved in the Rho/ROCK signaling pathway, essential for neuronal degeneration[22] and regeneration[23]. An additional 45 CpGs were identified at a more relaxed significance threshold of *P*-value $< 1 \times 10^{-5}$ (Fig. 1, Supplementary Data 2). For these 50 AD-associated CpGs, the odds ratios ranged from 0.833 to 1.156 in the ADNI dataset and 0.748 to 1.178 in the AIBL dataset. In the meta-analysis, the mean and median effect size of methylation beta values (i.e., parameter estimate for methylation beta

values in the logistic regression model) across all CpGs were −0.33 and −0.27, respectively. The mean and median effect size of methylation beta values for the 50 AD-associated CpGs were −3.20 and −6.00, respectively. The majority of the 50 significant CpGs were hypomethylated in AD subjects (35 CpGs), located outside CpG islands or shores (44 CpGs), or were in distal regions located greater than 2k bp from the TSS (41 CpGs). These observations are consistent with the knowledge that during aging, the strongest risk factor for AD, DNA methylation levels at intergenic regions are marked with hypomethylation[24]. Only a small number (9 CpGs) of these 50 CpGs were located in promoters of genes, which included *NR1I2*, *STMN2*, *TREML1*, *FKBP5*, *EIF2D*, *PXK*, *UPK3B*, *TDGF1*, and *RCCD1*.

Using meta-analysis *P*-values for individual CpGs as input, comb-p[25] software identified 9 differentially methylated regions (DMRs) at 5% Sidak *P*-value after multiple comparison corrections (Table 2). The number of CpGs in these DMRs ranged from 3 to 9. There was little overlap between the DMRs and the AD-associated CpGs; only 3 DMRs overlapped with the top 50 CpGs. Among the 9 DMRs, the majority (6 out of 9) were hypomethylated in AD. Five DMRs were in the promoter regions of the *PM20D1*, *NNAT*, *EIF2D/LGTN*, and *C1orf65* genes. The *PM20D1* gene is associated with response to accumulation of amyloid-β in AD brains[26,27], the *NNAT* gene is associated with neurodegeneration[28], and the *EIF2D* gene is critical for adaption to cellular stress in neurodegenerative diseases such as AD[29]. Also, blood methylation level at cg27202708 in the *C1orf65* DMR was one of 14 CpG sites previously shown to be associated with regional brain volumes measured by magnetic resonance imaging[30]. Interestingly, among these significant CpGs and DMRs, 20 CpGs and 3 DMRs were also located in enhancer regions (Supplementary Data 2, Table 2), which are regulatory DNA sequences that transcription factors bind to activate gene expressions[31,32]. Taken together, these results demonstrated the results of our meta-analysis are consistent with recent epigenomics literature in brain research. In addition to replicating methylation differences in genes previously implicated in AD (e.g., *PM20D1*), we also nominated additional genes associated with AD (e.g., *C1orf65*).

### Cross-tissue meta-analysis identified AD-associated DNA methylation differences in both brain and blood

To identify CpGs and genomic regions with DNA methylation differences in both brain and blood samples, we next performed a cross-tissue meta-analysis of six datasets by additionally including four brain prefrontal cortex (PFC) samples datasets, generated by the ROSMAP[2], Mt. Sinai[4], London[3], and Gasparoni[33] methylation studies, along with the two blood sample datasets described above. We previously meta-analyzed these four PFC brain datasets and identified a number of

**Table 1 | In epigenome-wide meta-analysis of blood DNA methylation in ADNI and AIBL datasets, 5 CpGs were significant in the Alzheimer's disease (AD) vs. cognitive normal groups comparison at 5% false discovery rate after multiple comparison correction**

| CpG | chr | pos | Meta-analysis | | | | | Annotations | |
| --- | --- | --- | --- | --- | --- | --- | --- | --- | --- |
| | | | OR | OR.CI | pValue | FDR | direction | GREAT | Illumina |
| cg03429569 | chr2 | 176,978,289 | 0.945 | (0.926,0.965) | 1.19E-07 | 3.68E-02 | -- | HOXD10 (−3018) | |
| cg14195992 | chr8 | 48,265,917 | 0.874 | (0.831,0.919) | 1.43E-07 | 3.68E-02 | -- | SPIDR (+92751);CEBPD (+385731) | SPIDR/ KIAA0146 |
| cg10570276 | chr20 | 36,810,754 | 1.108 | (1.067,1.152) | 1.53E-07 | 3.68E-02 | ++ | BPI (−121771);TGM2 (−16981) | |
| cg14727962 | chr5 | 178,784,230 | 0.903 | (0.869,0.938) | 2.22E-07 | 3.91E-02 | -- | RUFY1 (−193329);ADAMTS2 (−11800) | |
| cg03718411 | chr5 | 31,280,802 | 0.942 | (0.921,0.964) | 2.72E-07 | 3.91E-02 | -- | CDH6 (+86946);DROSHA (+251366) | CDH6 |

Inverse-variance weighted fixed-effects meta-analysis models were used to combine dataset-specific results from logistic regression models that included covariate variables age, sex, batch, and immune cell-type proportions. Odds ratios (OR) describe changes in odds of AD (on the multiplicative scale) associated with a one percent increase in methylation beta values (i.e., increase in methylation beta values by 0.01) after adjusting for covariate variables. Direction indicates hypermethylation (+) or hypomethylation (-) in AD samples compared to controls in the ADNI and AIBL datasets. All statistical tests are two-sided.

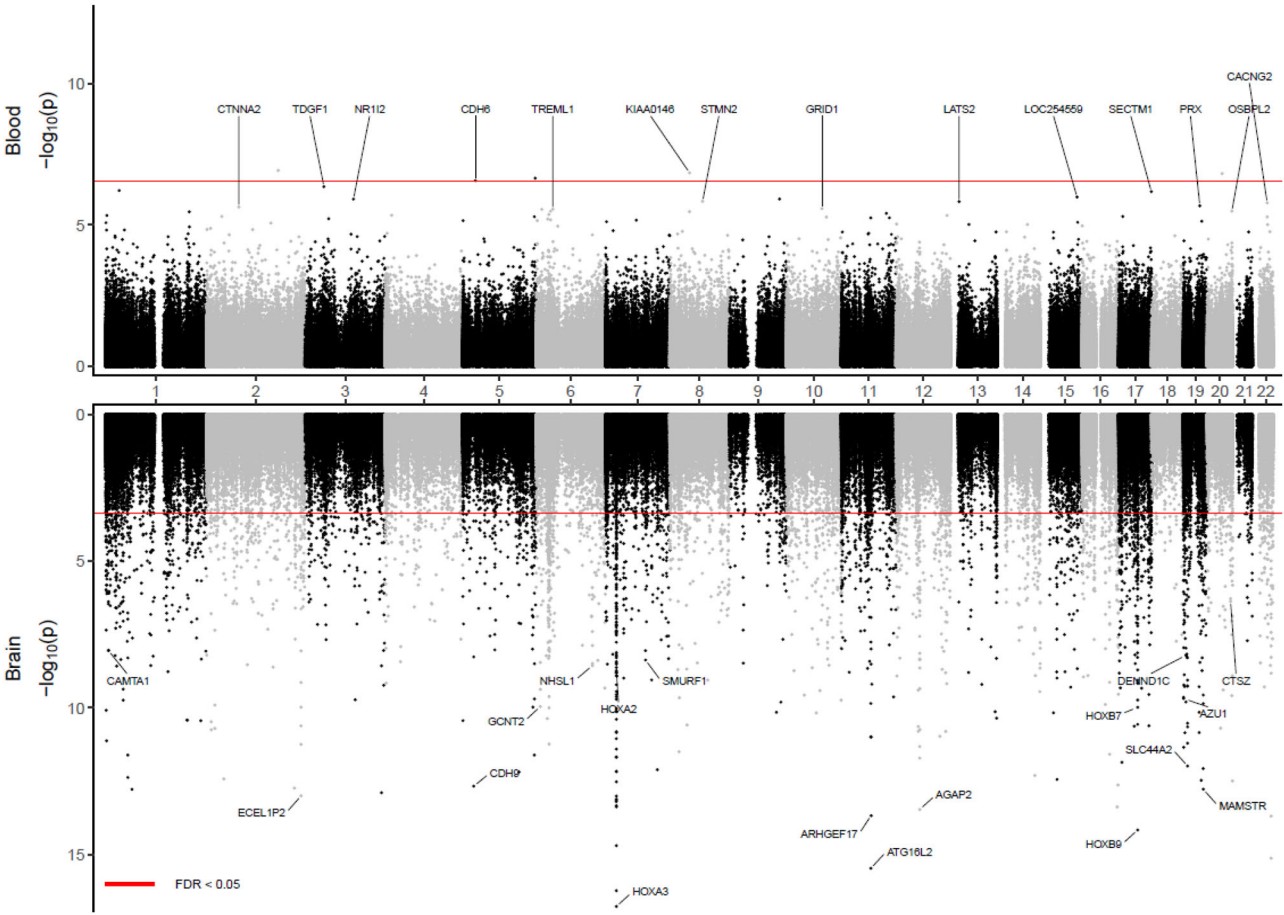

**Fig. 1 | Miami plot for epigenome-wide meta-analysis of blood DNA methylation in ADNI and AIBL datasets.** The X-axis shows chromosome numbers. The Y-axis shows $-\log_{10}(P\text{-value})$ of methylation to AD diagnosis association in the blood (above X-axis) or methylation to AD neuropathology (Braak stage) association in the brain (below X-axis). The genes corresponding to the top 20 most significant CpGs in blood or brain samples meta-analyses are highlighted. The P-values for brain sample meta-analysis were obtained from Zhang et al. (2020) (PMID: 33257653). The red line indicates a significance threshold of 5% False Discovery Rate for multiple comparison correction. All statistical tests are two-sided.

**Table 2 | In epigenome-wide meta-analysis of blood DNA methylation in ADNI and AIBL datasets, comb-p software identified 9 DMRs in the Alzheimer's disease (AD) vs. cognitive normal groups comparison at 5% Sidak P-values adjusted for multiple comparisons, which were then annotated using the GREAT software and the enhancer regions described in Nasser et al. (2021) study (PMID: 33828297)**

| DMR | nProbes | P-value | Sidak-P | Direction | GREAT Annotation | Enhancer |
|---|---|---|---|---|---|---|
| chr1:205819345-205819464 | 5 | 5.16E-11 | 3.12E-07 | -- | PM2OD1 (−160) | No |
| chr20:36148672-36148861 | 9 | 1.11E-10 | 4.22E-07 | -- | NNAT (−850) | Yes |
| chr1:206786170-206786181 | 3 | 1.67E-10 | 1.09E-05 | ++ | EIF2D (−272) | No |
| chr20:36149081-36149232 | 6 | 2.47E-10 | 1.18E-06 | --- | NNAT (−460) | Yes |
| chr13:21578684-21578734 | 3 | 8.32E-10 | 1.20E-05 | -- | XPO4 (−101760);LATS2 (+56977) | No |
| chr14:56777451-56777526 | 4 | 9.52E-10 | 9.13E-06 | ++ | TMEM260 (−269022);PELI2 (+192396) | No |
| chr7:2728841-2728913 | 3 | 4.07E-09 | 4.06E-05 | -- | AMZ1 (+9721);GNA12 (+155081) | Yes |
| chr1:202172848-202172913 | 4 | 2.71E-08 | 3.00E-04 | ++ | LGR6 (+9852);UBE2T (+138227) | No |
| chr1:223566643-223566710 | 5 | 3.73E-08 | 4.00E-04 | -- | C1orf65 (−38) | No |

Direction indicates hypermethylation (+) or hypomethylation (-) in AD samples compared to controls in the ADNI and AIBL datasets. All statistical tests are two-sided.

CpGs significantly associated with AD Braak stage[5], a standardized measure of neurofibrillary tangle burden determined at autopsy[34]. Supplementary Data 3 includes detailed information (e.g., Braak stage, clinical diagnosis, PMI) for the brain samples. At 5% FDR, our cross-tissue meta-analysis identified 365 CpGs and 40 DMRs. We then prioritized 97 CpGs and 10 DMRs by requiring these CpGs and DMRs to be also FDR-significant (i.e., FDR < 0.05) in our previous brain sample

meta-analysis[5] and at least nominally significant (i.e., P-value < 0.05) in our current blood meta-analysis (Fig. 2). Among these prioritized CpGs and DMRs, about half of them (52 CpGs and 6 DMRs) were located in enhancer regions[31] (Supplementary Data 4, Table 3). Also, the majority (74 CpGs and 8 DMRs) showed opposite directions of changes in the brain and blood. Among the top 10 most significant prioritized CpGs and DMRs, only a few methylation differences showed a consistent

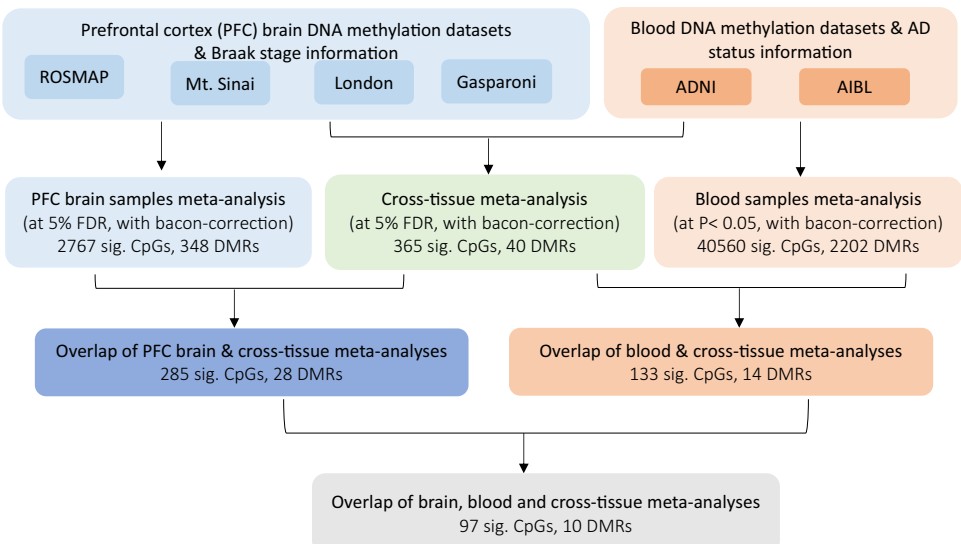

**Fig. 2 | Workflow for identifying cross-tissue DNA methylation differences that are associated with both AD pathology (in prefrontal cortex brain samples) and AD diagnosis (in blood samples).** Genomic corrections were performed using the bacon method (PMID: 28129774) in the analysis of all individual datasets. For brain sample meta-analysis, we obtained 3751 CpGs at 5% FDR, which reduced to 2767 CpGs after bacon correction.

direction of change in both tissues: one CpG (cg05157625) and one DMR (chr1:167090618-167090757) were mapped to gene body of the *RIN3* gene and intergenic region, respectively. A second DMR (chr17:62009607-62009835) was located in the promoter region of the *CD79B* gene (Table 3). Beyond PFC, additional brain regions in the cortex, such as temporal gyrus (TG) and entorhinal cortex (EC) are often also affected by neurodegeneration in AD. Smith et al. studied differential methylation in the cortex associated with AD Braak stage and identified 236, 95, and 10 significant CpGs (at 5% Bonferroni adjusted *P*-value) in the PFC, TG and EC, respectively[35]. Our cross-tissue analyses of the blood samples with brain regions in the TC and EC nominated significant DNA methylation differences at 8 CpGs and 1 CpG, respectively (Supplementary Figs. 2, 3, Supplementary Data 5). These CpGs and DMRs that are significant in both brain and blood tissues highlighted AD-associated DNA methylation differences in the periphery that are also altered in the brain.

**Integrative multi-omics analyses revealed functional implications of the DNA methylation differences in AD**
To better understand the functional roles of these significant DMRs and CpGs, we next performed several integrative analyses using matched DNA methylation and gene expression samples measured on the same subjects. More specifically, for blood sample analysis, we analyzed matched blood samples from 265 independent subjects (84 AD cases and 181 controls) in the ADNI study. For brain sample analysis, we analyzed matched samples measured on prefrontal cortex tissues of 529 subjects (428 AD cases and 101 controls) in the ROSMAP study.

**Correlation of methylation levels of significant CpGs and DMRs in AD with expressions of nearby genes.** We first correlated DNA methylation levels of the significant DMRs or CpGs with the expression levels of genes found in their vicinity. For the 50 CpGs that reached *P*-value <$10^{-5}$ in blood sample meta-analysis, after removing effects of covariate variables in both DNA methylation and gene expression levels separately (Methods), we found 2 CpGs (cg07886485 and cg23963071) located in promoter regions of the *PXK* and *SERPINB9* genes, respectively, were significantly associated with target gene expression levels at 5% FDR (Supplementary Data 6a, Supplementary Fig. 4). One DMR (chr1:205819345-205819464), also located in the promoter region, showed a strong negative association with the gene

expression level of the target gene *PM20D1* (*P*-value < $2.22 \times 10^{-16}$, FDR < $2.22 \times 10^{-16}$).

Among the 97 CpGs and 10 DMRs that were significant in the cross-tissue analysis of brain and blood samples (Fig. 2), no methylation-gene expression pairs reached 5% FDR in the ADNI dataset. In the ROSMAP[36] brain sample dataset, after removing covariate effects in DNA methylation and gene expression separately, 10 CpGs and one DMR were significantly associated with their target genes (Supplementary Data 6b). Among them, DNA methylations at 7 CpGs and 1 DMR, located on *MYO1F*, *RIN3*, *HOXA7*, *T*, *PRAM1*, *CSDC2*, *L3MBTL2*, and *THBS1* genes, showed the same direction of association with target gene expressions in both brain and blood samples. The greater number of methylation to gene expression associations detected in brain samples compared to blood samples could be due to the larger sample size of matched methylation-RNA brain samples available (529 ROSMAP brain samples vs. 265 ADNI blood samples).

In this section, we identified CpGs and DMRs that influence target gene expression directly. In the next section, we discuss identifying CpG methylations that influence target gene expression indirectly by modulating transcription factor activity.

**MethReg integrative analysis associated CpGs with putative transcription factors and target genes.** Transcription factors (TFs) are proteins that bind to DNA to facilitate transcription. Recent studies demonstrated that CpG methylation-dependent transcriptional regulation is a widespread phenomenon[37–39]. In particular, the binding of TFs onto DNA can be affected by DNA methylation, and DNA methylation can also be altered by proteins associated with the TFs. To better understand the regulatory roles of the AD-associated CpGs, we next performed an integrative analysis of DNA methylation, gene expression, and TF binding data to prioritize CpG-TF-target gene triplets in which regulatory activities of the TFs on target gene expressions are most likely influenced by CpG methylations, using our recently developed MethReg R package[40]. To estimate TF activities, the R package GSVA[41], which assesses expression levels of transcriptional targets of the TF proteins, was used.

At 5% FDR, our analysis of the ADNI blood samples identified a total of 8 CpG-TF-target gene triplets (Supplementary Data 7, Supplementary Figs. 5–7). The methylation-sensitive TFs in these triplets were involved in various biological processes previously implicated in AD, such as TGFβ signaling (SMAD3)[42], lipid metabolism (SREBF1)[43],

**Table 3 | Top 10 CpGs and DMRs prioritized in cross-tissue analysis, which are FDR-significant (i.e., FDR < 0.05 after multiple comparison correction) in both cross-tissue and brain samples meta-analyses, and nominally significant (i.e., P value < 0.05) in blood samples meta-analysis**

| CpG or DMR | chr | pos | meta-analysis P-values | | | FDR in cross tissue | P-values in brain datasets | | | | | P-values in blood datasets | | | Annotations | |
|---|---|---|---|---|---|---|---|---|---|---|---|---|---|---|---|---|
| | | | brain | blood | cross tissue | | GASPARONI | LONDON | MTSINAI | ROSMAP | direction | ADNI | AIBL | direction | GREAT | Enhancer |
| **CpGs** | | | | | | | | | | | | | | | | |
| cg25840926 | chr2 | 20647987 | 3.18E-10 | 5.44E-04 | 2.48E-10 | 9.04E-05 | 2.37E-01 | 1.51E-04 | 1.12E-02 | 2.06E-05 | ++++ | 1.95E-02 | 1.07E-02 | -- | RHOB (+1153);HS1BP3 (+202862) | Yes |
| cg14103343 | chr19 | 49220223 | 2.65E-12 | 3.68E-02 | 4.94E-10 | 9.04E-05 | 2.21E-01 | 7.31E-03 | 5.44E-05 | 6.56E-08 | ++++ | 2.80E-01 | 4.74E-02 | -- | MAMSTR (+2755);FUT2 (+20992) | No |
| cg12234455 | chr19 | 49220235 | 1.40E-11 | 1.59E-02 | 9.01E-10 | 1.10E-04 | 1.61E-01 | 4.91E-03 | 4.15E-05 | 2.03E-06 | ++++ | 1.50E-01 | 4.38E-02 | -- | MAMSTR (+2743);FUT2 (+21004) | No |
| cg18137450 | chr1 | 224620806 | 1.42E-07 | 6.82E-04 | 2.59E-09 | 1.41E-04 | 8.32E-02 | 9.10E-02 | 3.97E-04 | 2.09E-04 | ++++ | 1.10E-01 | 5.46E-04 | -- | WDR26 (+1195);CNIH4 (+76255) | No |
| cg23859635 | chr2 | 42795262 | 1.75E-11 | 7.57E-03 | 7.01E-09 | 2.57E-04 | 1.09E-01 | 2.96E-06 | 3.54E-04 | 1.03E-03 | ++++ | 1.96E-01 | 9.25E-03 | -- | MTA3 (-395) | No |
| cg14019523 | chr14 | 94407033 | 2.07E-09 | 6.58E-03 | 1.46E-08 | 4.88E-04 | 4.16E-01 | 1.19E-02 | 3.38E-03 | 5.17E-06 | ---- | 3.83E-02 | 7.66E-02 | ++ | PRIMA1 (-152207);ASB2 (+36104) | No |
| cg05157625 | chr14 | 93153553 | 9.79E-12 | 3.81E-02 | 2.14E-08 | 6.04E-04 | 1.30E-01 | 8.50E-05 | 6.60E-03 | 3.25E-06 | ++++ | 1.53E-02 | 5.25E-01 | ++ | LGMN (+6147);RIN3 (+173436) | No |
| cg13374901 | chr20 | 60639404 | 1.94E-09 | 4.13E-02 | 6.14E-08 | 1.25E-03 | 3.64E-01 | 6.52E-04 | 1.46E-01 | 2.85E-07 | ++++ | 4.37E-01 | 2.01E-02 | -- | TAF4 (+1462);CDH4 (+811923) | No |
| cg00682096 | chr17 | 46672924 | 2.92E-09 | 4.72E-03 | 7.23E-08 | 1.33E-03 | 4.42E-02 | 2.07E-04 | 1.36E-03 | 2.21E-03 | ++++ | 2.11E-02 | 1.01E-01 | -- | HOXB5 (-1602) | No |
| cg14622996 | chr6 | 32109801 | 3.69E-07 | 9.74E-04 | 7.99E-08 | 1.33E-03 | 7.05E-01 | 1.09E-03 | 2.32E-02 | 5.74E-04 | ---- | 8.09E-02 | 2.30E-03 | ++ | FKBPL (-11734);PRRT1 (+9928) | No |
| **DMRs** | | | | | | | | | | | | | | | | |
| chr16:57405979-57406511 | | | 3.20E-07 | 8.22E-04 | 4.98E-06 | 3.24E-03 | 6.30E-01 | 2.38E-04 | 2.63E-01 | 1.83E-03 | ++++ | 2.52E-03 | 9.49E-02 | -- | CX3CL1 (-125) | No |
| chr6:31554829-31555016 | | | 5.24E-07 | 3.27E-03 | 6.13E-06 | 3.24E-03 | 5.23E-01 | 2.07E-03 | 1.06E-02 | 1.96E-03 | ---- | 4.84E-03 | 2.64E-01 | ++ | LTB (-472l);LST1 (-54) | Yes |
| chr6:30853948-30854233 | | | 4.72E-08 | 2.72E-02 | 6.46E-06 | 3.24E-03 | 5.40E-01 | 3.57E-02 | 1.12E-03 | 2.55E-04 | ++++ | 2.88E-01 | 4.52E-02 | -- | GTF2H4 (-21870);DDR1 (+2230) | Yes |
| chr15:39871808-39872186 | | | 3.67E-08 | 6.11E-03 | 2.29E-05 | 1.00E-02 | 4.83E-01 | 7.53E-04 | 5.25E-04 | 3.16E-02 | ++++ | 3.11E-01 | 6.02E-03 | -- | THBS1 (-1297) | Yes |
| chr6:166876490-166877038 | | | 1.11E-04 | 1.35E-03 | 3.15E-05 | 1.23E-02 | 4.26E-01 | 7.59E-03 | 1.01E-01 | 1.37E-02 | ---- | 3.97E-03 | 1.06E-01 | ++ | MPC1 (-80278);RPS6KA2 (+399275) | Yes |
| chr1:167090618-167090757 | | | 2.78E-06 | 3.50E-02 | 8.84E-05 | 2.22E-02 | 3.23E-01 | 1.74E-02 | 3.95E-03 | 4.28E-03 | ---- | 5.96E-02 | 3.22E-01 | -- | POU2F1 (-99378);DUSP27 (+27406) | Yes |
| chr17:7832680-7832943 | | | 1.56E-06 | 3.67E-02 | 1.23E-04 | 2.54E-02 | 3.73E-01 | 6.27E-02 | 3.51E-03 | 5.11E-03 | ++++ | 1.01E-02 | 5.32E-01 | -- | CNTROB (-2661);KCNAB3 (-59) | No |
| chr17:62009607-62009835 | | | 9.99E-07 | 3.52E-02 | 1.55E-04 | 2.71E-02 | 8.62E-01 | 5.31E-04 | 1.93E-03 | 1.90E-02 | ---- | 3.28E-01 | 1.95E-02 | -- | CD79B (-25) | No |
| chr12:125028166-125028339 | | | 5.45E-06 | 1.14E-02 | 2.27E-04 | 3.06E-02 | 4.96E-01 | 1.46E-02 | 1.41E-01 | 2.63E-03 | ++++ | 5.74E-03 | 5.22E-01 | -- | NCOR2 (-48455);SCARB1 (+320140) | Yes |
| chr10:682693-682871 | | | 8.85E-05 | 1.74E-02 | 3.42E-04 | 3.94E-02 | 2.33E-02 | 2.32E-01 | 1.13E-01 | 2.85E-02 | ---- | 9.71E-03 | 4.83E-01 | ++ | DIP2C (+52824);ZMYND11 (+502358) | No |

The brain samples meta-analysis results were obtained from Zhang et al. (2020) (PMID: 33257653). In brain and blood samples meta-analyses, inverse-variance weighted meta-analysis models were applied to the brain and blood sample datasets separately. In the cross-tissue meta-analysis, Stouffer's method was used to combine weighted z-scores (transformed from P-values) in all six datasets, where the weights were specified based on the square root of the total number of subjects in each dataset. For region-based analysis, because DMRs identified with comb-p vary by datasets so cannot be meta-analyzed, we used coMethDMR to analyze DMRs at a common set of genomic regions across different datasets. The CpGs and DMRs were annotated using the GREAT software and the enhancer regions described in Nasser et al. (2021) study (PMID: 33828297). Direction indicates hypermethylation (+) or hypomethylation (-) in AD samples compared to controls in the individual datasets. The CpG and DMRs with the same direction of effect on AD pathology in the brain and AD diagnosis in the blood are shown in Italic. All statistical tests are two-sided.

inflammatory response (CEBPB, XBP1), and cell cycle control (MYC). Similarly, at 5% FDR, our analysis of ROSMAP brain samples identified 9 CpG-TF-target gene triplets (Supplementary Data 8, Supplementary Figs. 8–11). The methylation-sensitive TFs in these triplets included several critical regulators in AD, such as the estrogen receptor (ESR1)[44] and the glucocorticoid receptor (NR3C1)[45], as well as factors in biological pathways previously shown to be important in AD pathogenesis, such as antioxidant response (MAFF)[46,47], TGFβ signaling (SMAD1)[42], Wnt signaling (TCF7L2)[48], and hypoxia (H1F1A)[49].

One example is the triplet cg16908123-RUNX3-C1orf100 in the analysis of the ADNI blood samples, in which DNA methylation at cg16908123 appeared to attenuate the repression of target gene *C1orf100* by TF RUNX3 (Supplementary Fig. 5, Supplementary Data 7c). Notably, in samples with low cg16908123 methylation levels, higher TF activities corresponded to more repression of the target gene (*P*-value = $1.18 \times 10^{-5}$). On the other hand, when methylation is high, the target gene is relatively independent of TF activities. Therefore, methylation at cg16908123 and the TF RUNX3 jointly regulate *C1orf100* gene expression. In contrast, individually, neither methylation at CpG cg16908123 nor TF activities for RUNX3 were associated with gene expression of the target gene *C1orf100*.

The transcription factor RUNX3 is a critical regulator for linage specificity of hematopoietic stem and progenitor cells, which ensures a balanced output of peripheral blood cell types. It has been observed that during aging, RUNX3 level decreases, accompanied by shifts in blood cell types[50]. Previously, several studies also demonstrated RUNX3 activity is primarily regulated by DNA methylation[51–53]. The target gene *C1orf100* was recently discovered to be associated with white cell telomere length[54]. Although the role of *C1orf100* is not well understood in AD, given shortening of telomere length is a hallmark of aging, DNA methylation differences and transcription factor activities that affect this gene are particularly relevant for AD. In both ADNI and AIBL studies, methylations at cg16908123 were significantly hypermethylated in AD samples (Supplementary Data 4), consistent with the MethReg prediction that CpG methylation attenuates the effect of RUNX3 and the previous observations that RUNX3 activity decreases with aging.

Our MethReg analysis revealed a number of additional TFs that interact with AD-associated CpG methylations to jointly regulate target gene expressions (Supplementary Data 7, 8). Importantly, for these methylation-sensitive TFs, the TF-target associations are often only present in a subset of samples with high (or low) methylation levels, thus might be missed by analyses that use all samples (Supplementary Figs. 5–11). Although many of the TFs have previously been implicated to AD, our integrative analysis provided additional information on the specific roles of the TFs in transcriptional regulation, and identified target genes for these TFs in AD, by nominating plausible TF-target gene associations that are mediated by DNA methylations.

**Integrative analysis revealed gene expressions associated with DNA methylation differences in the blood and the brain converge in biological pathways.** To further understand the functional effects of the significant CpGs as a group, we next performed an additional integrative analysis[55] (Supplementary Fig. 12). Covariate effects were first removed from DNA methylation and gene expression data by fitting separate linear models (Methods). We next performed a principal component analysis. For each sample, we summarized covariate-adjusted methylation values in the significant CpGs by the first PC (PC1), which is a weighted linear combination of the methylation values; these are the methylation PC scores (MPS). Then, we tested the association between the MPS and covariate-adjusted genome-wide gene expression levels using linear models, ranked the genes by the absolute value of t-statistics for MPS, and performed pathway analysis using the GSEA method[56,57]. This analysis was performed for ADNI blood samples and ROSMAP brain samples separately.

At 5% FDR, we identified 1864, 80 genes, and 13 genes that were significantly associated with the methylation PC scores in the brain, blood, or both tissues, respectively (Supplementary Data 9). A substantial number of the 13 genes significant in both tissues included genes involved in the inflammatory response, such as *CD79A, LY86, SP100, CD163, CD200*, and *MS4A1*, recapitulating the prominence of immune processes in AD[58]. For GSEA pathway analysis, at 5% FDR, 830, 12, and 10 Canonical Pathways (CP) were significant in the analyses of the brain, blood, or both tissues (Supplementary Data 10). Similarly, 1895, 52, and 47 Gene Ontology (GO) terms were significant in the analysis of brain, blood, or both tissues (Supplementary Data 11). Notably, the number of significant GO terms and pathways overlapping in analyses of brain and blood was significantly more than expected by chance (CP: *P*-value = 0.019, GO: *P*-value = $4.50 \times 10^{-10}$).

Among the pathways that reached 5% FDR significance in brain or blood analyses, a number of pathways were involved in inflammatory responses in AD, such as neutrophil degranulation, antigen processing and presentation, interferon signaling, and activation of nuclear factor kappa B pathways. Additional significant biological processes included biological processes previously shown to be important in AD, such as glycolysis, antiviral mechanism, endocytosis, mRNA translation[59,60], and retrograde transport[61]. Importantly, some of these biological processes also pointed to potential biomarkers and therapeutic targets for the treatment of AD. For example, the Rho GTPase signaling pathway, the second most significant pathway in brain samples analysis (*P*-value = $3.62 \times 10^{-21}$, FDR = $3.28 \times 10^{-18}$), plays critical roles in regulating synaptic plasticity of the neurons and has been studied as a viable target for AD[62–64]. Also, within the ATM signaling pathway, which was significant in blood sample analysis (*P*-value = $1.19 \times 10^{-4}$, FDR = 0.026), the ATM protein is a central regulator for DNA damage response and has been proposed as a promising new target for treating neurodegeneration recently[65,66].

A total of 10 canonical pathways and 47 GO terms reached 5% FDR in both brain and blood sample analyses (Supplementary Data 10b, 11c). To further compare the genes that contributed most to the enrichment scores in these pathways[56], we next estimated the Jaccard similarity coefficients (percentage of overlapping genes) for leading-edge genes, which most strongly contributed to pathway enrichment, in each pathway in the analyses of brain and blood samples. Among all pathways that were significant in both tissues, the Jaccard coefficients ranged from 0.19 to 0.36 for canonical pathways and 0.22 to 0.42 for GO BP terms (Supplementary Data 10b, 11c), indicating only a moderate percentage of overlapping leading-edge genes in the same pathway across different tissues. Interestingly, neutrophil degranulation ranked as the most significant pathway in both brain and blood sample analyses. However, for this pathway, only 29.5% of leading-edge genes were shared in brain and blood sample analyses (Supplementary Data 10c). These results suggested there might be a convergence in pathways across the brain and blood in gene expression changes associated with the methylation PC scores.

**Correlation and overlap with genetic susceptibility loci**
To identify methylation quantitative trait loci (mQTLs) for the significant DMRs and CpGs, we next performed look-up analyses using the GoDMC database[67] for mQTLs in blood and the xQTL sever[68] for mQTLs in the brain. Among the CpGs mapped to the 50 AD-associated CpGs (Supplementary Data 2) or located in AD-associated DMRs in blood samples analysis (Table 2), 39 CpGs had mQTLs in *cis*, and 7 CpGs had mQTLs in *trans*. Similarly, among the 97 prioritized CpGs or CpGs located in the 10 prioritized DMRs in our cross-tissue analysis (Fig. 2), 13, 104, and 11 CpGs had mQTLs in the brain, blood, or both, respectively. Among them, 72 CpG – mQTL pairs were significant in both brain and blood sample analysis (Supplementary Data 12). The larger number of mQTLs detected in the blood could be due to the

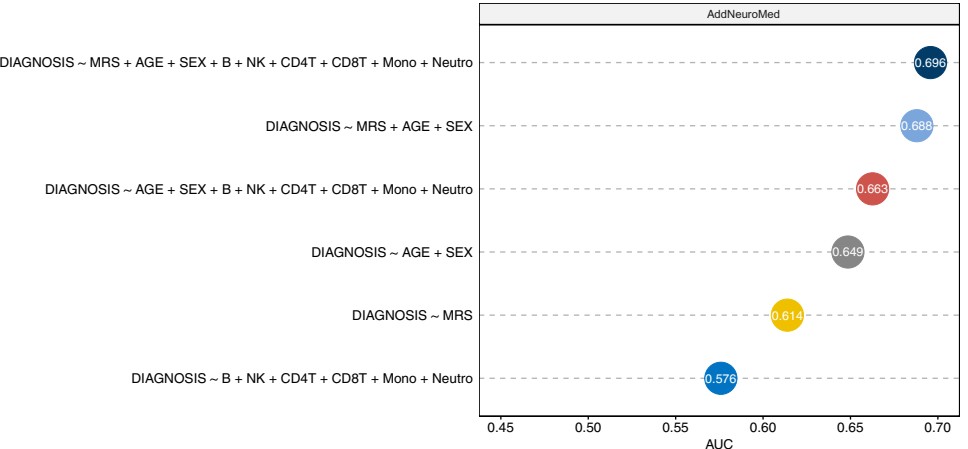

**Fig. 3 | Performance of different logistic regression models for predicting AD diagnosis in out-of-sample validation.** The training samples included 135 AD cases and 356 control samples from the AIBL dataset, and the testing samples included 83 AD cases and 88 control samples from the AddNeuroMed dataset. Among the 97 prioritized CpGs in cross-tissue analysis, 91 CpGs are also available in the testing dataset (i.e., AddNeuroMed). Methylation risk score (MRS) was computed as the sum of methylation beta values for these 91 CpGs weighted by their estimated effect sizes in the AIBL dataset. Abbreviation: AUC Area Under ROC curve, MRS methylation risk score.

larger sample size of GoDMC, which used a meta-analysis design[67], compared to xQTL, which was computed based on a single dataset (ROSMAP)[68].

Next, to evaluate if the significant mQTLs in the brain and blood overlapped with genetic risk loci implicated in AD, we compared the mQTLs with the 24 LD blocks of genetic variants reaching genome-wide significance in a recent meta-analysis of AD GWAS[69]. While no brain mQTLs overlapped with the 24 LD blocks, 3045 blood mQTLs were mapped to the LD region chr6:32395036-32636434, which included genetic variants mapped to the *HLA-DRA, HLA-DRB5, HLA-DRB1, HLA-DQA1*, and *HLA-DQB1* genes (Supplementary Data 13). In addition, we also evaluated if the significant methylation differences overlapped with the genetic risk loci implicated in AD. We found only 3 AD-associated CpGs overlapped with genetic variants mapped to the *TREM2, SPI1*, and *ACE* genes (Supplementary Data 14), and no DMRs overlapped with any of the 24 LD blocks. These results are consistent with another recent study that meta-analyzed 11 blood-based EWAS of neurodegenerative disorders, including AD, amyotrophic lateral sclerosis, and Parkinson's disease, which also did not find evidence for overlap between significant EWAS loci and GWAS loci[16]. The lack of commonality between genetic and epigenetic loci in AD could be due to a lack of power in EWAS and/or GWAS but could also reflect the relatively independent roles of genetic variants and DNA methylation in influencing AD susceptibility[70,71].

### Out-of-sample validations of AD-associated DNA methylation differences in an external dataset

To evaluate the feasibility of the identified methylation differences for predicting AD diagnosis, we next performed out-of-sample validations using an external DNA methylation dataset generated by the AddNeuroMed study, which included 83 AD cases and 88 control samples with ages greater than 65 years[10]. To this end, we computed the methylation risk scores (MRS)[72], which were shown to have excellent discrimination of smoking status, and moderate discrimination of obesity, alcohol consumption, HDL cholesterol[73], and amyotrophic lateral sclerosis case-control status[74] recently.

More specifically, for each sample in the AddNeuroMed dataset, we computed MRS by summing the methylation beta values of the significant CpGs weighted by their estimated effect sizes in the AIBL dataset. Among the 97 significant CpGs in cross-tissue analysis, 91 CpGs are available in the AddNeuroMed dataset and were used for computing MRS. Several logistic regression models were estimated using

the AIBL dataset and then tested on the AddNeuroMed dataset. We considered logistic regression models with three sources of variations that might affect the prediction for AD diagnosis: known clinical factors (i.e., age and sex), estimated cell-type proportions for each sample, and MRS. When tested individually, the models that included age and sex, cell types, or MRS alone had AUCs of 0.649. 0.576, and 0.614, respectively (Fig. 3). When combined with estimated cell types or MRS, prediction performance for the model with clinical factors (i.e., age and sex) improved to AUCs of 0.663 and 0.688, respectively. Notably, MRS was computed based on fewer than one hundred CpGs, while the six variables corresponding to cell type proportions were estimated using all CpGs on the array. The best performing model included age, sex, MRS, and cell types (AUC = 0.696, 95% CI: 0.616 − 0.770) (Figs. 3 and 4), significantly more predictive than a random classifier with an AUC of 0.5 (P-value = 2.78 × 10⁻⁵).

The same logistic regression model trained using both ADNI and AIBL datasets (instead of AIBL alone) performed slightly worse with an AUC of 0.678, which might be due to batch effect of different training datasets. The model that included MRS based on AD-associated CpGs (instead of prioritized CpGs) also performed worse with an AUC of 0.609, probably because CpGs with cross-tissue differences also leveraged information from additional brain sample datasets. In addition to MRS, we also evaluated the performance of MPS (methylation PC scores) described above, which sums methylation beta values in the testing dataset weighed by loadings of the first principal component in PCA analysis. The best performing logistic regression model involving MPS was also estimated using the AIBL dataset, included variables age, sex, cell types, and MPS computed based on the same 91 prioritized cross-tissue CpGs, and achieved an AUC of 0.662.

## Discussion

We performed a comprehensive meta-analysis of two large independent AD blood samples EWAS, which generated DNA methylation profiles using the same Infinium MethylationEPIC BeadChip. After correcting for multiple comparisons, a total of 5 CpGs reached 5% FDR (Table 1), and an additional 45 CpGs reached P-value < 1 × 10⁻⁵ (Supplementary Data 2). Among them, two CpGs (cg03546163 and cg14195992), mapped to the *FKBP5* and *SPIDR* genes, also reached genome-wide significance in another large meta-analysis of multiple EWAS (of Amyotrophic lateral sclerosis, Parkinson's disease, and AD)[16], suggesting these two CpGs corresponded to susceptibility loci common in neurodegenerative diseases. In mouse models of AD, *FKBP5*

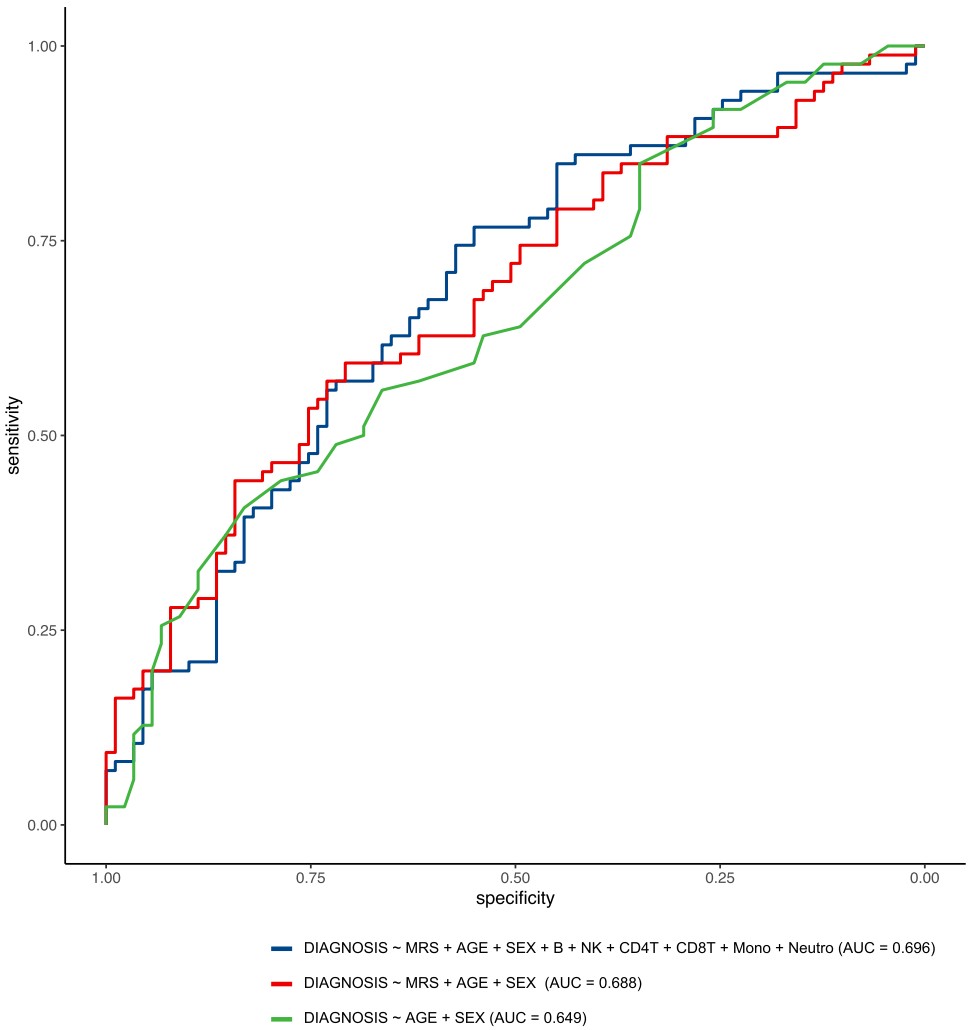

DIAGNOSIS ~ MRS + AGE + SEX + B + NK + CD4T + CD8T + Mono + Neutro (AUC = 0.696)

DIAGNOSIS ~ MRS + AGE + SEX  (AUC = 0.688)

DIAGNOSIS ~ AGE + SEX (AUC = 0.649)

**Fig. 4 | Receiver Operating Characteristic curves (ROCs) for logistic regression models predicting AD diagnosis in out-of-sample validation using the AddNeuroMed dataset (83 AD cases and 88 controls).** The training dataset included 135 AD cases and 356 control samples from the AIBL dataset. The best performing logistic regression model (with AUC = 0.696) included methylation risk score (MRS), age, sex, and estimated cell-type proportions, where MRS was computed as the sum of methylation beta values for 91 prioritized CpGs in cross-tissue analysis weighted by their estimated effect sizes in the AIBL dataset.

was shown to promote tau protein aggregation and is primarily regulated by DNA methylation[75]. It has been observed that *FKBP5* expression levels increase with age and are upregulated in AD brains[75]. Our results and those from Nabais et al. (2021) provided additional evidence that DNA methylation at cg03546163 is also significantly hypomethylated in the blood of AD subjects[16], suggesting *FKBP5* methylation might be a viable candidate biomarker for tau pathology. Moreover, genetic variants on *FKBP5* have also been implicated in stress-related disorders such as major depressive disorder[76,77], suicide behavior[77], posttraumatic stress disorder[78], and childhood maltreatment[79]. These results are consistent with the increased dementia risk observed in patients with psychiatric disorders such as depression[80].

Our most significant DMR is located on the *PM20D1* gene, where highly significant promoter hypomethylation was observed in AD patients ($P$-value = $5.16 \times 10^{-11}$, Sidak adjusted $P$-value = $3.12 \times 10^{-7}$). In addition, the strong negative methylation-gene expression correlation at this locus was the strongest association we observed in our study ($P$-value < $2.2 \times 10^{-16}$, FDR < $2.2 \times 10^{-16}$). Previously, promoter hypermethylation of the *PM20D1* gene, which is associated with responses to the neurotoxic insults in AD brains, has been consistently observed in multiple AD brain samples EWAS[27]. Moreover, it was shown that overexpression of *PM20D1* is associated with decreased amyloid-β

levels and reduces cell death both in vitro and in vivo; thus, it may have a neuroprotective role against AD[26,27].

The second most significant DMR is located in the promoter region of the *NNAT* gene, which encodes the neuronatin protein that aggregates and causes cell death in another neurodegenerative disease called Lafora disease[28]. The *NNAT* gene is mainly expressed in the brain during neurogenesis and is critical for maintaining synaptic plasticity of the neurons[81]. Interestingly, *NNAT* is an imprinted gene and normally only expresses the paternal allele, while the maternal allele is suppressed by DNA methylation. Changes in neuronatin might be a common downstream effect due to neuronal loss in multiple diseases such as Lafora disease, diabetes, and cancer[28]. Importantly, DNA methylation at the *NNAT* gene is diet responsive and can be altered by food enriched with methyl group donors, including folate, choline, methionine, and vitamin B12[82]. Future studies are needed to evaluate the effect of methyl donor diet on AD.

Consistent with observations from individual EWAS studies[3,83], the CpGs and DMRs identified in this meta-analysis of AD blood samples EWAS were mostly distinct from the significant CpGs previously identified in our meta-analysis of AD brain samples EWAS[5]. Among the AD-associated CpGs and DMRs, only one CpG, cg06357748 mapped to the *RAD52* gene, was significant in both studies (i.e., it reached 5% FDR in brain samples meta-analysis[5] and $P$-value < $10^{-5}$ in the current blood

samples meta-analysis). Interestingly, we also found that a group of *cis*-mQTLs was strongly associated with cg06357748 in both brain and blood with effects in the same direction (Supplementary Data 12). RAD52 is a critical protein in the RNA-templated recombination repair pathway that mends oxidative damage to DNA in neurons, and it was recently shown that RAD52 activities could be inhibited by high concentrations of amyloid β, which may lead to neuronal genomic instability and neurodegeneration[84]. Using the same significance threshold (5% FDR), we detected much fewer significant CpGs (5 CpGs) in our current blood samples meta-analysis than in the previous brain samples meta-analysis. This is probably due to the larger between-samples variability in blood samples. For example, in the London dataset[3], which had brain and blood DNA methylation levels measured on the same subjects, we observed the average standard deviation for methylation M-values[85] across all CpGs in the blood samples is 0.303 in healthy controls, compared to 0.265 for brain samples from the same subjects.

Moreover, in contrast to results from our brain samples meta-analysis[5], in which the majority of methylation differences were located in CpG islands and hyper-methylated in AD, we found the majority of methylation differences in our blood samples meta-analysis were located in open seas and hypo-methylated in AD. A number of other recent studies also observed hypomethylated CpGs in the blood of AD patients. For example, Fransquet et al. (2020)[8] observed the majority of differentially methylated CpGs between dementia cases at diagnosis and controls in their study also had lower methylation levels in the cases; Madrid et al. (2018)[12] discovered hypomethylated CpGs in the *B3GALT4* and *ZADH2* genes in patients with late-onset AD; Mitsumori et al. (2020)[13] identified and replicated lower DNA methylation levels in CpG island shores of *CR1*, *CLU*, and *PICALM* genes in the blood of Japanese AD patients.

Our results are also consistent with previous studies on aging, the strongest risk factor for AD. DNA methylation changes throughout the lifetime, and it has been observed that as people age, methylation decreases in intergenic regions but increases at many promoter-associated CpGs islands regions[86]. For example, a comparison of DNA methylation in CD4 + T cells of a centenarian with a newborn revealed pervasive hypomethylation across the genome[24] in the aged individual. Similarly, Reynolds et al. (2014)[87] studied age-related CpGs that are associated with gene expression (age-eMS) in human monocytes and T cells and found age-eMS tended to be hypomethylated as people become older. These previous observations of hypomethylation during aging, combined with our observation of hypomethylation in AD subjects compared to cognitively normal subjects of similar ages (Supplementary Data 1), are consistent with the hypothesis that some age-associated DNA methylation changes are accentuated by AD[88–90].

While it is difficult to infer whether epigenetic variations are a cause or consequence of the disease process, the DNA methylation differences we observed in this study can provide a useful source of candidate markers associated with neuronal and axonal cell injuries in AD. A number of the differentially methylated genes corresponded to gene expression and protein biomarkers already being explored for diagnosing or tracking progressions in AD. For example, CSF levels of the CDH6 protein (Table 1) were recently shown to be significantly associated with CSF p-tau and t-tau levels as well as amyloid-beta in a large cohort of dementia patients[21]. Our second most significant CpG, cg14195992, which also reached genome-wide significance in Nabis et al. (2020)[16], is mapped to the gene body of the *SPIDR* gene. Previously, gene expression at the *SPIDR* gene was among one of 48 selected genes for a classifier that discriminated AD cases from control samples[91].

This study has several limitations. First, the methylation levels analyzed here were measured on whole blood, which contains a complex mixture of cell types. To reduce confounding effects due to different cell types, we included estimated cell-type proportions as covariate variables in all our analyses. After multiple comparison corrections using FDR, we only identified a few significant associations between AD-associated CpGs or DMRs with expression levels of nearby genes, this lack of association might be due to larger variability in the samples introduced by cellular heterogeneity. Future studies that utilize single-cell technology for gene expression and DNA methylation might improve power and shed more light on the particular cell types affected by the AD-associated DNA methylation differences discovered in this study. Second, in the analysis of blood samples, we used clinical diagnosis for AD as our endpoint. However, as the pathophysiological process of AD can begin many years before the onset of clinical symptoms[92,93], there can be disagreement between neuropathology and clinical phenotypes. Our analysis may have included individuals with disparate brain pathology and clinical diagnosis. These individuals would likely have diluted DNA methylation to AD association signals in our study; therefore, our meta-analysis results may be conservative. Future studies that utilize in vivo neuroimaging endophenotypes that measure amyloid and tau might improve the power of blood-based DNA methylation studies. Third, in the meta-analysis of blood samples, only 5 CpGs reached 5% FDR, we therefore examined 45 additional CpGs at the less stringent significance threshold of $P$-value $< 10^{-5}$. Also, to select DNA methylation differences that are significant in the brain, blood, and cross-tissue meta-analyses (Fig. 2), instead of using the FDR-significant CpGs in blood samples meta-analysis, we intersected nominally significant (i.e., $P$-value $< 0.05$) CpGs and DMRs in blood sample meta-analysis with FDR-significant DNA methylation differences in brain and cross-tissue meta-analyses. These more relaxed significance thresholds might correspond to higher false-positive rates. To help prioritize the most biologically relevant DNA methylation differences, we performed several integrative analyses of methylation with genetic variants, gene expressions, and transcription factor binding sites. Fourth, we did not consider MCI subjects in this study because there is considerable heterogeneity among MCI subjects, with subjects converting to AD at different trajectories. As ADNI is currently conducting additional phases of the study, future analyses with a larger sample size will make it possible to detect more DNA methylation differences in AD as well as in MCI subjects. Fifth, the MRS-based risk prediction model could also be further improved. Because DNA methylation samples in the testing dataset (AddNeuroMed) were measured by 450k arrays, which are different from the EPIC arrays used by the AIBL study, we only included CpGs that mapped to both arrays in the computation of our MRS. The performance of our MRS-based prediction models can be assessed more accurately using future testing dataset measured by the EPIC arrays. Also, in the out-of-sample validation analysis, we did not include other important factors such as APOE genotype, which might also significantly predict AD diagnosis[94] because we did not have access to APOE information in the AIBL and AddNeuroMed datasets. Our internal validation using the ADNI dataset suggested additionally including APOE (ε4 allele) into our best-performing logistic regression model, which included MRS, age, sex, and estimated cell types (Figs. 3, 4), could substantially improve prediction performance. More specifically, a 10-fold cross-validation using the ADNI dataset showed the estimated average AUCs for the best performing logistic regression models with and without APOE status were 0.691 and 0.810, respectively (Supplementary Data 16). Finally, the associations we identified do not necessarily reflect causal relationships. Additional studies are needed to establish the causality of the nominated DNA methylation markers.

Although brain tissues are ideal for studying AD, currently, it still is not feasible to obtain methylation levels in brain tissues from living human subjects. On the other hand, because of the relative ease of obtaining blood samples, measuring blood methylation levels is a practical alternative. In this study, we identified a number of DNA methylation differences consistently associated with AD diagnosis in

blood samples of two large independent datasets of subjects. Our integrative analysis of DNA methylation differences in the blood with those in the brain, as well as gene expression and TF binding sites information prioritized a number of CpGs, genes, pathways, and regulators that are associated with both neuropathology and/or AD diagnosis, many of which were involved in the inflammatory responses in AD. Consistent with previous studies, we found the patterns of DNA methylation differences in the brain and blood resemble those observed during aging. Given advanced age is the greatest risk factor for AD, our results highlight the need for a better understanding of epigenetic changes during normal aging to design prevention and treatment strategies for AD[88]. Despite the limited agreement at significant individual CpGs across tissues, we did find that expression changes associated with the combined DNA methylation differences (methylations summarized by principal component score) in brain or blood were enriched in a number of common pathways, suggesting systems biology approaches[95] that uncover the pathways disrupted in AD subjects might be a useful strategy for identifying peripheral AD biomarkers that reflect changes in the brain. With respect to genetic variants, consistent with previous studies[70,71], we also observed AD-associated DNA methylation differences are mostly independent of genetic effects, suggesting multi-omics models[96] that leverage complementary information from both the genome and the epigenome would be helpful for predicting AD risk. Finally, given the relatively modest sample size of our training dataset, the significant discriminatory classification of AD samples with our MRS-based risk prediction model demonstrated DNA methylation might be a predictive biomarker for AD. Future studies that validate our findings in larger and more diverse community-based cohorts are warranted.

## Methods

### Study datasets

Our meta-analysis included a total of 1284 whole blood samples from two independent datasets, generated by the ADNI[14] and AIBL[16] studies. The external validation samples included 171 whole blood samples generated by the cross-European AddNeuroMed study[10]. DNA methylation samples for the ADNI, AIBL, and AddNeuroMed studies were obtained from adni.loni.usc.edu, Gene Expression Omnibus (GEO) (accession: GSE153712 and GSE144858), respectively. Only samples older than 65 years were analyzed.

### Preprocessing of DNA methylation data

All DNAm samples were measured by the same Illumina Human-Methylation EPIC beadchip, which included more than 850,000 CpGs[97]. Supplementary Data 15 shows the number of CpGs and samples removed at each quality control step. More specifically, quality control for CpG probes included several steps: first, we selected probes with a detection $P$-value < 0.01 for all the samples in the dataset. A small detection $P$-value corresponds to a significant difference between signals in the probes compared to background noise. Next, using function rmSNPandCH from the DMRcate R package (version 2.4.1), we removed probes that are located on the X and Y chromosomes, are cross-reactive[98], located close to single nucleotide polymorphism (SNPs) (i.e., an SNP with minor allele frequency (MAF) ≥ 0.01 was present in the last five base pairs of the probe), or are associated with cigarette smoking[99].

Quality control for samples included restricting our analysis to samples with good bisulfite conversion efficiency (i.e., ≥85%). In addition, principal component analysis (PCA) was used to exclude outlier samples. To this end, PCA was performed using the 50,000 most variable CpGs for each dataset, and samples within ±3 standard deviations from the mean of PC1 and PC2 were selected to be included in the final sample set. For the ADNI dataset, we also selected one sample among multiple technical replicate samples and removed samples without slide information or matched clinical data.

The quality-controlled methylation datasets were next subjected to the QN.BMIQ normalization procedure as recommended by a recent systematic study of different normalization methods[100]. More specifically, we first applied quantile normalization as implemented in the lumi R package (version 2.42.0) to remove systematic effects between samples. Next, we applied the β-mixture quantile normalization (BMIQ) procedure as implemented in the wateRmelon R package (version 1.34.0) to normalize beta values of type 1 and type 2 design probes in the Illumina arrays.

### Blood sample meta-analysis

For the AIBL dataset, the association between CpG methylations and diagnosis (dementia vs. cognitive normal or CN) was assessed using logistic regression models with logit (probability of dementia) as the outcome, methylations beta values as the main independent variable, and covariate variables age, sex, methylation plate, and estimated cell-type proportions (B lymphocytes, natural killer cells, CD4 + T lymphocytes, CD8 + T lymphocytes, monocytes, neutrophils). For the analysis of ADNI dataset, which is a longitudinal study with some subjects contributing multiple observations, we applied logistic mixed-effects models that additionally included subjects random effects to account for correlations from multiple observations generated from the same subjects.

More specifically, cell type proportions were estimated using the EpiDISH[101] R package (version 2.6.0). For the AIBL dataset, logistic regression models were fitted using the glm() function in R software (version 4.0.3). For the ADNI dataset, logistic mixed-effects models were fitted using Procedure GLMMIX in SAS software (version 9.4). In the AIBL dataset, because age information was not available, sample ages were estimated using the DNAm-based-age-predictor[102] (https://github.com/qzhang314/DNAm-based-age-predictor/, elastic net method). For ADNI samples, age was calculated as the difference between the date on which blood was drawn and the birthdate of the subject.

We estimated genomic inflation factors (lambda values) using both the conventional approach[103] and the *bacon* method[104], which is specifically proposed for a more accurate assessment of inflation in EWAS. Briefly, the bacon method uses a Bayesian algorithm to estimate a three-component normal mixture given the observed test statistics (e.g., t-statistics corresponding to the effect of DNA methylation in regression models) where one component reflects the null distribution, and two other components correspond to the positive and negative associations in the data. Mean and standard deviations of the estimated (empirical) null distribution correspond to bias and inflation of the test statistics. The lambda values ($\lambda$) by the conventional approach were 0.54 and 1.24, and lambdas based on the bacon approach ($\lambda$.bacon) were 0.73 and 0.96 for the ADNI and AIBL datasets, respectively.

Next, for more accurate statistical assessment, genomic correction using the bacon method[104], as implemented in the bacon R package, was applied to obtain bacon-corrected effect sizes, standard errors, and $P$-values for each dataset. Efron (2010) showed that in large-scale simultaneous testing situations (e.g., when many CpGs are tested in an analysis), serious defects in the theoretical null distribution may become obvious, while empirical Bayes methods can provide much more realistic null distributions[105]. By definition, the bacon-corrected test statistics have estimated an inflation factor of 1 because empirical null distributions were used in their estimation. Indeed, after bacon-correction, the estimated inflation factors were $\lambda$ = 0.97 and 1.02, and $\lambda$.bacon = 0.98 and 0.97, for the ADNI and AIBL datasets, respectively.

To meta-analyze individual CpG results across different datasets, we used the inverse-variance weighted fixed-effects model, which was implemented in the meta R package (version 4.18.0). The estimated effect sizes from the meta-analysis of logistic regression model results were then exponentiated to compute odds ratios for a one percent increase in beta values (i.e., increase in beta values by 0.01).

For the region-based meta-analysis, we used the comb-p method[25]. Briefly, comb-p takes single CpG P-values and locations of CpG sites to scan the genome for regions enriched with a series of adjacent low P-values. In our analysis, we used meta-analysis P-values of the two whole blood sample datasets as input for comb-p. As comb-p uses the Sidak method to account for multiple comparisons, we considered DMRs with Sidak P-values less than 0.05 to be significant. We used the default setting for our comb-p analysis, with parameters -seed 1e-3 and -dist 200, which required a P-value of $10^{-3}$ to start a region and extend the region if another P-value was within 200 base pairs.

**Cross-tissue meta-analysis** Our cross-tissue meta-analysis included the 1284 blood samples from the ADNI ($n = 793$) and AIBL ($n = 491$) datasets described above, and an additional 1030 prefrontal cortex brain samples from four independent datasets, which included samples from the ROSMAP ($n = 726$), Mt. Sinai ($n = 141$), London ($n = 107$), and Gasparoni ($n = 56$) studies that we previously analyzed in our brain samples meta-analysis[5]. We used Stouffer's Method[106], as implemented in sumz() function of R package metap, to combine weighted z-scores (transformed from P-values) in these six datasets. For each study, weights were specified based on the square root of the total number of subjects in each study[107]. For region-based analysis, because DMRs identified with comb-p often vary by datasets so cannot be meta-analyzed, the coMethDMR[108] R package (https://github.com/TransBioInfoLab/coMethDMR) was used to analyze a common set of genomic regions across datasets. For the cross-tissue meta-analysis of blood samples with other brain regions in the temporal gyrus and entorhinal cortex, we combined dataset-specific P-values of the CpGs in AIBL and ADNI with dataset-specific P-values for the CpGs in Supplementary Data 3, 5 of Smith et al.[35] using Stouffer's Method[106] as described above.

## Functional annotation of significant methylation differences

The significant methylation differences at individual CpGs and DMRs were annotated using both the Illumina (UCSC) gene annotation and GREAT (Genomic Regions Enrichment of Annotations Tool) software[109] that associates genomic regions to target genes. With the default "Basal plus method", GREAT links each gene to a regulatory region consisting of a basal domain that extends 5 kb upstream and 1 kb downstream from its transcription start site and an extension up to the basal regulatory region of the nearest upstream and downstream genes within 1 Mb. To assess the overlap between our significant CpGs and DMRs (CpG or DMR location +/− 250 bp) with enhancers, we used enhancer–gene maps generated from 131 human cell types and tissues described in Nasser et al. (2021)[31], available at https://www.engreitzlab.org/resources/. More specifically, we selected enhancer-gene pairs with "positive" predictions from the ABC model, which included only expressed target genes, does not include promoter elements, and has an ABC score higher than 0.015. In addition, we also required the enhancer-gene pairs were identified in cell lines relevant to this study (https://github.com/TransBioInfoLab/AD-meta-analysis-blood/blob/main/code/annotations/).

## Correlations between methylation levels of significant CpGs and DMRs in AD with expressions of nearby genes

For blood sample analysis, to evaluate the DNA methylation effect on the gene expression of nearby genes, we analyzed matched gene expression (Affymetrix Human Genome U 219 array) and DNA methylation (EPIC array) data from 265 independent subjects in the ADNI study. We considered the 97 prioritized CpGs (Supplementary Data 4), the 50 CpGs that achieved P-value < $1 \times 10^{-5}$ in AD vs. CN analysis (Supplementary Data 2), as well as CpGs located in the 10 prioritized DMRs (Table 3) and 9 significant DMRs in the AD vs. CN comparison (Table 2).

To associate genes with DNA methylation sites, we used the MethReg R package[40] and considered CpGs located in the promoter regions and distal regions separately. More specifically, for CpGs located in the promoter region (within ±2 kb around the transcription start sites (TSS)), we tested the association between CpG methylation with expression levels of the target genes. On the other hand, for CpGs in the distal regions (>2 kb from TSS), we tested the association between CpG methylation with expression levels of ten nearest genes upstream and downstream from the CpG. For gene expression data, when multiple probes were mapped to a gene, we used the median gene expression level over all probes mapped to the gene as its gene expression level.

To reduce the effect of potential confounding effects, when testing for methylation to gene expression associations, we first adjusted for age at visit, sex, immune cell-type proportions (for B lymphocytes, natural killer cells, CD4 + T lymphocytes, CD8 + T lymphocytes, monocytes, neutrophils), and batch effects in both DNA methylation and gene expression levels separately and extracted residuals from the linear models. Immune cell-type proportions were estimated using the R packages EpiDISH[101] and Xcell[110] (https://github.com/dviraran/xCell) for DNA methylation and gene expression data, respectively. A separate robust linear model was then used to test the association between methylation residuals and gene expression residuals, adjusting for AD status. The analysis of DMRs was performed similarly, except by replacing CpG methylation levels with the median methylation level of all CpGs located within the DMR.

For the analysis of brain samples, we considered the 97 prioritized CpGs (Supplementary Data 4) and CpGs within ours 10 prioritized DMRs (Table 3) in cross-tissue analysis, and performed similar analyses using matched RNA-seq and DNA methylation brain samples from 529 independent subjects in the ROSMAP study. First, we removed confounding effects in DNA methylation data by fitting the linear model *DNA methylation M value ~ neuron.proportions + batch + sample.plate array + ageAtDeath + sex* and extracting residuals from this model, which are the methylation residuals. The proportions of neurons in each sample were estimated using the CETS R package[111]. Similarly, we also removed potential confounding effects in RNA-seq data by fitting model *log2(normalized FPKM values + 1) ~ age at death + sex + markers for cell types* and extracting gene expression residuals. The last term, "markers for cell types," included multiple covariate variables to adjust for the multiple types of cells in the brain samples. More specifically, we used the expression levels of genes that are specific for the five main cell types present in the central nervous system[2]: *ENO2* for neurons, *GFAP* for astrocytes, *CD68* for microglia, *OLIG2* for oligodendrocytes, and *CD34* for endothelial cells, and included these as variables in the above linear regression model.

We then tested for association between methylation residuals and gene expression residuals, adjusting for the Braak stage using a separate robust linear model *residuals_{expression} ~ residuals_{DNAm} + Braak stage*. The analysis of DMRs was performed similarly, except by replacing CpG methylation levels with the median methylation level of all CpGs located within the DMR.

## MethReg integrative analysis

To create the CpG-TF-target gene triplets, we first linked a given CpG to transcription factors (TFs) with binding sites within ±250 bp of the CpG, by using information from the ReMap2020 database[112], which contains regulatory regions for over one thousand transcriptional regulators obtained using genome-wide DNA-binding experiments such as ChIP-seq. Next, we linked a CpG to a gene if the CpG was within its promoter region; otherwise, we considered the CpG to be in the distal regions (>2k bp from any promoter regions) and linked it to 5 genes upstream and 5 genes downstream of the CpG location. The CpG-TF pairs are then combined with CpG-target gene pairs to create triplets of CpG-TF-target genes. We used the same 265 and 529 matched methylation-RNA samples from ADNI and ROSMAP studies described in the section above for blood and brain samples analysis.

Methylation residuals and gene expression residuals were obtained in the same way as described above and used as input for the MethReg analysis. TF activities were estimated using the GSVA[41] R package. The MethReg analyses were performed using the MethReg R package[40].

## Integrative analysis of DNA methylation differences in the brain and blood with transcriptome-wide gene expressions

Covariates adjusted DNA methylation levels (i.e., DNA methylation residuals described above) at AD-associated CpGs in blood samples or Braak-associated CpGs in brain samples were summarized using principal component analysis (PCA) (Supplementary Fig. 12). More specifically, for blood sample analysis, the first principal component (PC1) was computed by performing PCA analysis on the ADNI dataset with the 50 CpGs that achieved $P$-value $< 10^{-5}$ in AD vs. CN analysis (Supplementary Data 2); while for the brain sample analysis, PCA was performed on the ROSMAP dataset, and PC1 was computed for the 3751 CpGs identified in our previous brain sample meta-analysis[5]. For each sample, PC1 was estimated using the prcomp() function in R, and they represented the methylation PC scores (MPS).

Next, we tested the association between log-transformed gene expressions and the methylation PC scores using linear regression models, adjusting for age, sex, batch, and estimated immune cell types in brain and blood samples separately. Gene Set Enrichment Analysis[56] was performed using the fgsea R package (version 1.17.1). As a ranking measure for each gene, we used the absolute value of t-statistics corresponding to methylation PC scores in the linear model, which associated gene expression levels with the methylation PC scores. We analyzed gene sets from the Molecular Signatures Database (MSigDB)[56], accessed by the msigdbr R package (version 7.4.1.), including the GO Biological Process terms (C5:BP) and Canonical pathways (BIOCARTA, KEGG, REACTOME, WikiPathways) (C2:CP). To evaluate if the number of significant GO terms and pathways in the analyses of brain and blood samples was significantly more than expected by chance, we used Fisher's exact test.

## Correlation and overlap with genetic susceptibility loci

We searched mQTLs using the GoDMC database[67] and the xQTL sever[68], which were downloaded from http://mqtldb.godmc.org.uk/downloads and http://mostafavilab.stat.ubc.ca/xQTLServe/, respectively. To select significant blood mQTLs in GoDMC, we used the same criteria as the original study[67], that is, considering a cis $P$-value smaller than $10^{-8}$ and a trans $P$-value smaller than $10^{-14}$ as significant. The 24 LD blocks of genetic variants reaching genome-wide significance were obtained from Supplementary 8 of Kunkle et al. (2019)[69].

## Out-of-sample validation

The best-performing risk prediction model was trained using samples from the AIBL dataset and tested on the AddNeuroMed dataset. More specifically, first, we computed the Methylation Risk Scores (MRS) as the sum of methylation beta values for prioritized CpGs in cross-tissue analysis (Supplementary Data 4) weighted by their estimated effect sizes (i.e., parameter estimate for methylation beta values in logistic regression after bacon-correction) in the AIBL dataset[72]. We included 91 prioritized CpGs that were available in the AddNeuroMed dataset from GEO. Next, the logistic regression model logit (Pr (AD)) ~ MRS + age + sex + B + NK + CD4T + CD8T + Mono + Neutro was fitted to the AIBL dataset using glm() function, and predict.glm() was used to apply the logistic regression model to AddNeuroMed dataset. The last six variables in the logistic regression model correspond to estimated proportions of different blood cell types of B-cells, Natural Killer (NK) cells, CD4 + T-cells, CD8 + T-cells, Monocytes, and Neutrophils, obtained using the EpiDish R package[101]. The R package pROC was used to estimate receiver operating characteristic curves (ROCs) and area under the ROC curves (AUCs). Similarly, logistic regression models with a subset of the variables in the above model (e.g., only age and

sex) were similarly developed using the AIBL dataset and tested on the AddNeuroMed dataset. To determine if a logistic regression model predicted AD diagnosis significantly better than chance, we used the Wilcoxon rank-sum test to compare estimated probabilities for AD cases versus controls[113].

## Internal validation to assess the impact of APOE genotype

Among the three public datasets (AIBL, AddNeuroMed, ADNI) we analyzed, the APOE genotype was only available for the ADNI dataset. To assess the added prediction accuracy of APOE gene, we performed internal validations (i.e., 10-fold cross-validations) using the ADNI dataset, by comparing our best performing model logit (Pr (AD)) ~ MRS + age + sex + B + NK + CD4T + CD8T + Mono + Neutro with the model that additionally included APOE ($\varepsilon$4 allele genotype). To obtain an independent set of samples, only the last visit of each subject in the ADNI dataset was used for this analysis. The function createFolds() in caret R package was used to divide the data into ten folds. Average AUCs over the ten iterations in the 10-fold cross-validations for the models with and without APOE were then estimated and compared.

In all analyses, to account for multiple comparisons, we computed FDR using the method of Benjamini and Hochberg[114]. Associations with 5% or less FDR were considered to be FDR significant. All analyses were performed using the R (https://www.r-project.org/; version 4.1.0), Python (version 2.7.12), SAS (version 9.4), and GREAT (version 4) software.

## Reporting summary

Further information on research design is available in the Nature Research Reporting Summary linked to this article.

## Data availability

All datasets analyzed in this study are publicly available as described in Methods. In particular, ADNI can be accessed from http://adni.loni.usc.edu, AIBL and AddNeuroMed datasets can be accessed from Gene Expression Omnibus (GEO) (accession: GSE153712, GSE144858). The Molecular Signatures Database (MSigDB) can be accessed from https://www.gsea-msigdb.org/gsea/msigdb/. The meta-analysis results generated by this study are available at https://github.com/TransBioInfoLab/AD-meta-analysis-blood.

## Code availability

The scripts for the analysis performed in this study can be accessed at https://github.com/TransBioInfoLab/AD-meta-analysis-blood/ and the zenodo repository (https://doi.org/10.5281/zenodo.6787981) .

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

## Acknowledgements

This research was supported by US National Institutes of Health grants R21AG060459 (L.W), R01AG061127 (L.W.), and R01AG062634 (E.R.M, L.W.). The ROSMAP study data were provided by the Rush Alzheimer's Disease Center, Rush University Medical Center, Chicago. Data collection for the ROSMAP dataset was supported through funding by NIA grants P30AG10161, R01AG15819, R01AG17917, R01AG30146, R01AG36836, U01AG32984, U01AG46152, the Illinois Department of Public Health, and the Translational Genomics Research Institute. Data used in preparation of this article were obtained from the Alzheimer's Disease Neuroimaging Initiative (ADNI) database (adni.loni.usc.edu). As such, the investigators within the ADNI contributed to the design and implementation of ADNI and/or provided data but did not participate in analysis or writing of this report. A complete listing of ADNI investigators can be found at: http://adni.loni.usc.edu/wp-content/uploads/how_to_apply/ADNI_Acknowledgement_List.pdf Data collection and sharing for the ADNI dataset was funded by the Alzheimer's Disease Neuroimaging Initiative (ADNI) (National Institutes of Health Grant U01 AG024904) and DOD ADNI (Department of Defense award number W81XWH-12-2-0012). ADNI is funded by the National Institute on Aging, the National Institute of Biomedical Imaging and Bioengineering, and through generous contributions from the following: AbbVie, Alzheimer's Association; Alzheimer's Drug Discovery Foundation; Araclon Biotech; BioClinica, Inc.; Biogen; Bristol-Myers Squibb Company; CereSpir, Inc.; Cogstate; Eisai Inc.; Elan Pharmaceuticals, Inc.; Eli Lilly and Company; EuroImmun; F. Hoffmann-La Roche Ltd and its affiliated company Genentech, Inc.; Fujirebio; GE Healthcare; IXICO Ltd.; Janssen Alzheimer Immunotherapy Research & Development, LLC.; Johnson & Johnson Pharmaceutical Research & Development LLC.; Lumosity; Lundbeck; Merck & Co., Inc.; Meso Scale Diagnostics, LLC.; NeuroRx Research; Neurotrack Technologies; Novartis Pharmaceuticals Corporation; Pfizer Inc.; Piramal Imaging; Servier; Takeda Pharmaceutical Company; and Transition Therapeutics. The Canadian Institutes of Health Research is providing funds to support ADNI clinical sites in Canada. Private sector contributions are facilitated by the Foundation for the National Institutes of Health (www.fnih.org). The grantee organization is the Northern California Institute for Research and Education, and the study is coordinated by the Alzheimer's Therapeutic Research Institute at the University of Southern California. ADNI data are disseminated by the Laboratory for Neuro Imaging at the University of Southern California.

## Author contributions

L.W., J.Y., E.R.M., T.C.S. designed the computational analysis. T.C.S., L.Z., L.G., M.A.S., A.V., L.W. analyzed the data. L.W., J.Y., E.R.M, T.C.S., X.C. contributed to interpretation of the results. T.C.S, L.W. wrote the paper, and all authors participated in the review and revision of the manuscript. L.W. conceived the original idea and supervised the project.

## Competing interests

The authors declare no competing interests.
