## [Peer Review File · Nature Communications]

Cross-tissue meta-analysis of blood and brain epigenome-wide association studies in Alzheimer's diseaseREVIEWER COMMENTS

Reviewer #1 (Remarks to the Author):

Silva et al. carried out a cross-tissue meta-analysis of blood and brain epigenome-wide association signals in Alzheimer's disease and complemented their findings with further analyses including gene expression and mQTL data.

The authors report a rich corpus of results. The manuscript is well written, but very dense. I believe the work has the potential to be relevant if the authors can make their results more easily accessible. Concretely, I have the following comments:

Two main points:

1) The aim of the work is unclear. Did the authors aim to establish a biomarker for AD, in which case I would probably like to see an out-of-sample validation or prediction performances. Also, for a biomarker it is not necessary to evidence cross-tissue concordance, unless the aim was also to identify mechanisms. But of what? Of risk or disease progression? In the introduction the authors write that cross-tissue biomarkers such 'provide objective information about changes in the underlying neuropathology in the ongoing disease process, which could be incredibly useful for monitoring the effects of disease-modifying treatment for neurodegenerative diseases'. Why did the authors then not use their methylation scores to predict clinical outcomes such as disease severity or disease progression? Did the longitudinal nature of ADNI not provide the opportunity to do so? Further down, the authors argue for 'biomarkers for [...] evaluating the effectiveness of candidate treatments'. Again, I would like to see some analyses around this statement.

Overall, I recommend the author to make their rationale clear and align their analysis with that rationale.

2) In large sections, the discussion reads like a repetition of the methods and results and could be replaced instead with an integration of findings into a coherent framework. What do we know now, why do we need to know and what can we do now? E.g. something like this:

a) We now have a list of CpGs that consistently link to AD in both blood and brain tissue. I'm less sure about what we can do with this now.

b) This list links also links to gene expression and more so in brain than in blood. But what does this tell us?

c) This list is not really influenced by genetic variants for AD. (there was an excellent discussion on this point in the results!)

So, overall, I encourage the authors to integrate the individual findings into one big picture and engage the reader into thinking about how to take this further.

Additional points:

3) The authors mention power as a challenge in this area of research. Although a post-hoc power analysis makes no sense, I was nevertheless missing a statement on effect sizes, variance explained or predictive potential of their findings. Also, how does the current study compare in terms of sample sizes to previous studies in blood and brain samples of AD (with the exception of their own previous study)?

4) I assume the red line in Figure 1 shows the cutoff of 10^{-7} ? Can the authors clarify?

5) The authors tend to use FDR as a cutoff with the notable exception for their first analyses in blood, where they use 10^{-5} or even 0.05. It looks like this was done because 'there was just no usable

findings at 10^{-7} or $-8'$. The authors mentioned this in passing in the discussion, but could be even more transparent about this and include a statement of the strengths and limitations of this approach.

6) Why was the data pruned for $>65y$?

7) I appreciate Figure 2 and encourage the authors to add more of those flow-chart figures to their manuscript to ease understanding. I could not find the supplementary figures.

8) I was confused by the fact that blood and brain methylation AD-related changes were opposite in direction yet the correlation between tissues were positive. How is this possible and what does this mean?

9) I was surprised by the lack of figures. I encourage the authors to evidence some of their findings with plots (which can often show the relevance of outliers or clustering). For example, SM Table 4a or 5 (and possibly many more) could be also visually presented.

10) Please add the FDR column to SM Table 4b - blood, unless there's a reason why it's missing.

11) The results section could benefit from the odd short summary at the end of each sub-section. For example, it seems that the methylation-expression relationship was stronger or at least more significant in brain compared to blood tissue. The end of the mQTL subsection provides an excellent example for this.

12) mQTLs: the authors find more mQTLs in blood than in brain tissue. Is that a function of sample size (i.e. is GoDMC much larger than xQTL able to detect smaller effects)?

13) Discussion: the authors mention consistent findings with Nabais et al. However, I believe AIBL data featured in both analyses, in which case consistency in findings should not be too surprising.

14) FKBP5 is >the< candidate gene for depression and childhood adversity. While the validity of candidate gene studies can be questioned, I feel that this deserved mentioning.

15) Re causality: I suppose the authors could carry out a Mendelian Randomization analysis to assess if the identified methylation markers might indeed be causal.

16) Methods: I was surprised to read that no age information was available in AIBL. I wonder what effect it has to control for epigenetically predicted age in an EWAS. Do the authors not risk losing potentially important epigenetic signals, especially since age itself is also strongly linked to the outcome?

17) The uncorrected lambda for ADNI were very low. Can the authors comment on that?

Reviewer #2 (Remarks to the Author):

The goal of this study was to identify blood-based DNA methylation markers that also change with underlying neuropathology in the brain and biomarkers for AD.

To do this, the authors profiled DNA methylation using the same Infinium MethylationEPIC BeadChip. They performed a cross-tissue meta-analysis by combining 1284 blood DNA methylation from two independent cohorts, generated by the ADNI and AIBL studies, and 1030 prefrontal cortex DNA methylation from four additional datasets.

One of the main findings is that expressions levels of 13 genes and 10 pathways were significantly associated with the AD-associated methylation differences in both brain and blood. A number of these

genes and pathways are involved in the immune responses in AD. Moreover, DNA methylation at cg05157625, located in RIN3 gene, is significantly associated with both AD diagnosis and neuropathology, and expression of the target gene in both tissues. The authors suggest cg05157625 could be a candidate biomarker for AD.

The overall approach is well performed, it meets the expected standards in the field, and it is novel as there are not that many studies in human samples studying this topic. The main findings support what is known, because they replicate methylation differences in genes previously implicated in AD, but also nominate additional genes that might be associated with AD. The approach is novel, performing integrative analysis of DNA methylation, gene expression, and TF (revealing that some TFs that might interact with AD-associated CpG methylations to jointly regulate target gene expressions). One of the main problems is the important overlap between aging and AD-associated methylation differences.

Specific comments:

- 1) The authors should include much more detail about the included samples. What is the diagnosis for those brains? Please include all the other variables relevant for this kind of sample: PMI, disease duration, Braak scores, CERAD, CDR, among others. I would suggest including other adjusting variables such as CDR, braak stage...
- 2) Analyzing the predictive value of these findings in an independent cohort would improve the manuscript.
- 3) It would have been interesting to integrate these results with different brain areas, not only prefrontal cortex.
- 4) How do you define "triplets"?

Reviewer #3 (Remarks to the Author):

The authors perform a comprehensive investigation of AD associated DNA methylation changes. First leveraging data by meta analysis in two large consortia for AD in blood, and then performing cross tissue analysis to assess relevance of the identified associations. Their approach yielded interesting results that match with the understanding of underlying AD etiology and add weight to previously identified genetic associations, while also supplying new genes of interest. mQTL analyses support cross tissue findings and identify interesting potential biomarkers. In general, this work is quite comprehensive and adds to the epigenetics and AD literature. There is very little the authors have not done to ensure robustness of their work and to assess that identified changes have functional or tissue specific relevance in the brain. Their findings match with and expand upon what is known about AD. Their discussion is well written and mentions the appropriate limitations inherent in methylation studies. I think this is a very solid piece of work and recommend publication.

Very minor comment:

You may wish to change the font from what appear to be pastes from the GO output tables into the text. This is not a reflection on the science or paper preparation in anyway, just something I noticed.

Minor comment:

The biomarker analysis identifying mQTL associated RIN3 with cross tissue methylation leads the authors to suggest it may be a good biomarker for AD. This suggestion is acknowledged as likely beyond scope for the current paper, which is packed with interesting analysis, but to further

investigate and support this biomarker assertion, it would be interesting to generate a model of RIN3 methylation for AD status and apply it to peripheral tissue samples in an independent cohort to assess its potential predictive efficacy through ROC and AUC metrics, etc. Of course, this would be most interesting in a high risk AD sample that had not yet developed the pathology and the availability of such samples is unknown to this reviewer. Still, such a biomarker would certainly be interesting, as current AD risk assessment at this time in my understanding just relies on APOE genetic assessment, scales, or potentially imaging and new biomarkers are needed to understand if novel prophylactic approaches like MAB therapy, for example, could be used in those at future risk.

Reviewer #1 (Remarks to the Author):

Silva et al. carried out a cross-tissue meta-analysis of blood and brain epigenome-wide association signals in Alzheimer's disease and complemented their findings with further analyses including gene expression and mQTL data. The authors report a rich corpus of results. The manuscript is well written, but very dense. I believe the work has the potential to be relevant if the authors can make their results more easily accessible.

We thank this reviewer for the enthusiasm and encouragement for our study. We have made substantial changes in response to this reviewer's helpful comments, which we will discuss in detail below.

Concretely, I have the following comments:

Two main points:

1) The aim of the work is unclear. Did the authors aim to establish a biomarker for AD, in which case I would probably like to see an out-of-sample validation or prediction performances. Also, for a biomarker it is not necessary to evidence cross-tissue concordance, unless the aim was also to identify mechanisms. But of what? Of risk or disease progression?

This reviewer raised an important point. We agree that a biomarker is not necessary to evidence cross-tissue concordance. To clarify, in this manuscript, our goals are to study DNAm differences in the blood from both mechanistic and biomarker perspectives. We're mainly interested in AD diagnosis, which is the main outcome of our blood samples meta-analysis.

In response to this reviewer's suggestion, we clarified our goals in the Introduction section. Please see (1) under "Revisions" below. As this reviewer suggested, we also added a section on out-of-sample validation to evaluate the prediction performance of the methylation risk scores in the Results section. Please see (2)-(5) under "Revisions" below.

In the introduction the authors write that cross-tissue biomarkers such 'provide objective information about changes in the underlying neuropathology in the ongoing disease process, which could be incredibly useful for monitoring the effects of disease-modifying treatment for neurodegenerative diseases'. Why did the authors then not use their methylation scores to predict clinical outcomes such as disease severity or disease progression? Did the longitudinal nature of ADNI not provide the opportunity to do so?

To clarify, because the ADNI dataset was used to identify CpGs with significant DNA methylation differences in our meta-analysis, to avoid overfitting, the ADNI dataset cannot also be used for prediction modeling. In response to this reviewer's comment, we performed prediction modeling using an independent dataset, the AddNeuroMed dataset (GEO accession: GSE144858); see details in item (2)-(5) under "Revisions" below.

Further down, the authors argue for 'biomarkers for [...] evaluating the effectiveness of candidate treatments'. Again, I would like to see some analyses around this statement.

In response, we have removed "evaluating effective of candidate treatments", and revised the last sentence in the Introduction. Please see (1) under "Revisions" below.

Overall, I recommend the author to make their rationale clear and align their analysis with that rationale.

We thank this reviewer for the good suggestion. In response, we have cleaned up the Introduction substantially by removing the 2nd paragraph and replaced it with a more focused paragraph that explains rationales for each analysis. Please see (1) under “Revisions” below.

Revisions

(1) Introduction

In this study, to help improve power, we meta-analyzed two large blood-based AD EWAS measured by the same Illumina Infinium MethylationEPIC BeadChips, and conducted by the Alzheimer’s Disease Neuroimaging Initiative (ADNI)¹ and the Australian Imaging, Biomarkers, and Lifestyle (AIBL)² consortiums recently. **We studied the AD-associated DNA methylation differences from both mechanistic and biomarker perspectives.** To this end, we also performed a cross-tissue meta-analysis by combining the DNA methylation datasets measured on blood with four additional DNA methylation datasets measured on over one thousand brain prefrontal cortex samples, to prioritize significant methylation differences associated with both AD diagnosis and AD neuropathology. **To understand functional roles of the methylation differences, we conducted several integrative analyses of methylations with genetic variants, gene expressions, and transcription factor binding sites.** In addition, we also evaluated the feasibility of the DNA methylation differences as potential biomarkers for diagnosing AD in an external blood samples dataset. Our analysis results provide a valuable resource for future mechanistic and biomarker studies in AD.

(2) In Results, added a new section on out-of-sample validation

Out-of-sample validations of AD-associated DNA methylation differences in an external cohort

To evaluate the feasibility of the identified methylation differences for predicting AD diagnosis, we next performed out-of-sample validations using an external DNA methylation dataset generated by the AddNeuroMed study, which included 83 AD cases and 88 control samples with ages greater than 65 years³. To this end, we computed the methylation risk scores (MRS)⁴, which were shown to have excellent discrimination of smoking status, and moderate discrimination of obesity, alcohol consumption, HDL cholesterol⁵, and amyotrophic lateral sclerosis case-control status⁶ recently.

More specifically, for each sample in the AddNeuroMed dataset, we computed MRS by summing the methylation beta values of the prioritized CpGs weighted by their estimated effect sizes in the AIBL dataset. Several logistic regression models were estimated using the AIBL dataset and then tested on the AddNeuroMed dataset. We considered logistic regression models with three sources of variations that might affect prediction for AD diagnosis: known clinical factors (i.e., age and sex), estimated cell-type

proportions for each sample, and MRS. When tested individually, the models that included age and sex, cell types, or MRS alone had AUCs of 0.649, 0.576, and 0.614, respectively (Figure 3). When combined with estimated cell types or MRS, prediction performance for the model with clinical factors (i.e., age and sex) improved to AUCs of 0.663 and 0.688, respectively. Notably, MRS was computed based on fewer than one hundred CpGs, while the six variables corresponding to cell type proportions were estimated using all CpGs on the array. The best performing model included age, sex, MRS, and cell types (AUC = 0.696, 95% CI: 0.616 - 0.770) (Figures 3-4), significantly more predictive than a random classifier with an AUC of 0.5 (P -value = 2.78×10^{-5}).

The same logistic regression model trained using both ADNI and AIBL datasets (instead of AIBL alone) performed slightly worse with an AUC of 0.678, which might be due to batch effect due to different training datasets. The model that included MRS based on AD-associated CpGs (instead of prioritized CpGs) also performed worse with an AUC of 0.609, probably because CpGs with cross-tissue differences also leveraged information from additional brain samples datasets. In addition to MRS, we also evaluated the performance of MPS (methylation PC scores) described above, which sums methylation beta values in the testing dataset weighed by loadings of the first principal component in PCA analysis. The best performing logistic regression model involving MPS was also estimated using the AIBL dataset, included variables age, sex, cell types, and MPS computed based on the same 91 prioritized cross-tissue CpGs, and achieved an AUC of 0.662.

(3) Added Figure 3 and Figure 4 for results of out-of-sample validation

Figure 3 Performance of different logistic regression models for predicting AD diagnosis in out-of-sample validation. The training samples included 135 AD cases and 356 control samples from the AIBL dataset and the testing samples included 83 AD cases and 88 control samples from the AddNeuromed dataset. MRS was computed as the sum of methylation beta values for 91 prioritized CpGs from cross-tissue analysis weighted by their estimated effect sizes in the AIBL dataset. Abbreviation: AUC = Area Under ROC curve

Figure 4 Receiver Operating Characteristic curves (ROCs) for logistic regression models predicting AD diagnosis in out-of-sample validation using the AddNeuroMed dataset (83 AD cases, 88 controls). The training dataset included 135 AD cases and 356 control samples from the AIBL dataset. The best performing logistic regression model (AUC = 0.696) included methylation risk score (MRS), age, sex, and estimated cell-type proportions, where MRS was computed as the sum of methylation beta values for prioritized CpGs weighted by their estimated effect sizes in the AIBL dataset.

(4) In Discussion, 2nd to the last paragraph, added text to discuss out-of-sample validation

This study has several limitations.... Fifth, the MRS-based risk prediction model could also be further improved. Because DNA methylation samples in the testing dataset (AddNeuroMed) were measured by 450k arrays, which are different from the EPIC arrays used by the AIBL study, we only included CpGs that mapped to both arrays in the computation of our MRS. The performance of our MRS-based prediction

models can be assessed more accurately using future testing dataset measured by the EPIC arrays. Also, in the out-of-sample validation analysis, we did not include other important factors such as APOE genotype, which might also significantly predict AD diagnosis⁷ because we did not have access to APOE information in the AIBL and AddNeuroMed datasets. Our internal validation using the ADNI dataset suggested additionally including APOE ($\epsilon 4$ allele) into our best performing logistic regression model, which included MRS, age, sex, and estimated cell types (Figures 3-4), might substantially improve prediction performance. More specifically, a 10-fold cross-validation using the ADNI dataset showed the estimated average AUCs for the best performing logistic regression models with and without APOE status were 0.691 and 0.810, respectively (Supplementary Table 16). Finally, the associations we identified do not necessarily reflect causal relationships. Additional studies are needed to establish the causality of the nominated DNA methylation markers.

(5) In Methods, last paragraph

Out-of-sample validation

The best-performing risk prediction model was trained using samples from the AIBL dataset and tested on the AddNeuroMed dataset. More specifically, first, we computed the Methylation Risk Scores (MRS) as the sum of methylation beta values (for prioritized CpGs in Supplementary Table 4) weighted by their estimated effect sizes (i.e., parameter estimate for methylation beta values in logistic regression after bacon-correction) in the AIBL dataset⁴. We included 91 prioritized CpGs that were available in the AddNeuroMed dataset from GEO. Next, the logistic regression model $\text{logit}(\text{Pr}(\text{AD})) \sim \text{MRS} + \text{age} + \text{sex} + \text{B} + \text{NK} + \text{CD4T} + \text{CD8T} + \text{Mono} + \text{Neutro}$ was fitted to the AIBL dataset using `glm()` function, and `predict.glm()` was used to apply the logistic regression model to AddNeuroMed dataset. The last six variables in the logistic regression model correspond to estimated proportions of different blood cell types of B-cells, Natural Killer (NK) cells, CD4+ T-cells, CD8+ T-cells, Monocytes, and Neutrophils, obtained using the EpiDish R package⁸. The R package pROC was used to estimate receiver operating characteristic curves (ROCs) and area under the ROC curves (AUCs). Similarly, logistic regression models with a subset of the variables in the above model (e.g., only age and sex) were similarly developed using the AIBL dataset and tested on the AddNeuroMed dataset. To determine if a logistic regression model predicted AD diagnosis significantly better than chance, we used the Wilcoxon rank-sum test to compare estimated probabilities for AD cases versus controls⁹. To assess the added prediction accuracy of APOE gene, we performed internal validations (i.e., 10-fold cross-validations) that compared our best performing model $\text{logit}(\text{Pr}(\text{AD})) \sim \text{MRS} + \text{age} + \text{sex} + \text{B} + \text{NK} + \text{CD4T} + \text{CD8T} + \text{Mono} + \text{Neutro}$ with the model that additionally included APOE ($\epsilon 4$ allele genotype) using the ADNI dataset. To obtain an independent set of samples, only the last visit of each

subject in the ADNI dataset was used for this analysis. The function `createFolds()` in `caret` R package was used to divide the data into 10 folds. Average AUCs over the 10 iterations in the 10-fold cross-validations for the models with and without APOE were then estimated and compared.

2) In large sections, the discussion reads like a repetition of the methods and results and could be replaced instead with an integration of findings into a coherent framework. What do we know now, why do we need to know and what can we do now? E.g. something like this:

a) We now have a list of CpGs that consistently link to AD in both blood and brain tissue. I'm less sure about what we can do with this now.

b) This list also links to gene expression and more so in brain than in blood. But what does this tell us?

c) This list is not really influenced by genetic variants for AD. (there was an excellent discussion on this point in the results!)

So, overall, I encourage the authors to integrate the individual findings into one big picture and engage the reader into thinking about how to take this further.

We thank this reviewer for the excellent suggestion. In response, we have shortened the Discussion section considerably by moving several paragraphs to Results. In addition, we also augmented the summary paragraph as the reviewer has suggested, to highlight the rationale, the big picture about our findings, and future directions.

Revision

(1) In Discussion

- Removed several paragraphs and moved them to the Results section.

(2) In Discussion, last paragraph

Although brain tissues are ideal for studying AD, currently, it is still not feasible to obtain methylation levels in brain tissues from living human subjects. On the other hand, because of the relative ease of obtaining blood samples, measuring blood methylation levels is a practical alternative. In this study, we identified a number of DNA methylation differences consistently associated with AD diagnosis in blood samples of two large independent cohorts of subjects. Our integrative analysis of DNA methylation differences in the blood with those in the brain, as well as gene expression and TF binding sites information prioritized a number of CpGs, genes, pathways, and regulators that are associated with both neuropathology and/or AD diagnosis, many of which were involved in the inflammatory responses in AD. Consistent with previous studies, we found the patterns of DNA methylation differences in the brain and blood resemble those observed during aging. Given advanced age is the greatest risk factor for AD, our results highlight the need for better understanding epigenetic changes during normal aging to design prevention and

treatment strategies for AD¹⁰. Despite the limited agreement at significant individual CpGs across tissues, we did find that expression changes associated with the combined DNA methylation differences (methylations summarized by principal component score) in brain or blood were enriched in a number of common pathways, suggesting systems biology approaches¹¹ that uncover the pathways disrupted in AD subjects might be a useful strategy for identifying peripheral AD biomarkers that reflect changes in the brain. With respect to genetic variants, consistent with previous studies^{12, 13}, we also observed AD-associated DNA methylation differences are mostly independent of genetic effects, suggesting multi-omics models that leverage complementary information from both the genome and the epigenome would be helpful for predicting AD risk. Finally, given the relatively modest sample size of our training dataset, the significant discriminatory classification of AD samples with our MRS-based risk prediction model demonstrated DNA methylation might be a predictive biomarker for AD. Future studies that validate our findings in larger and more diverse community-based cohorts are warranted.

Additional points:

3) The authors mention power as a challenge in this area of research. Although a post-hoc power analysis makes no sense, I was nevertheless missing a statement on effect sizes, variance explained or predictive potential of their findings. Also, how does the current study compare in terms of sample sizes to previous studies in blood and brain samples of AD (with the exception of their own previous study)?

This reviewer raised important points. In response, we added a statement on estimated effect sizes of methylation beta values in the meta-analysis of logistic regression models that associated them with AD diagnosis. Please see “Revision” below.

Although the coefficient of determination (R^2), which measures the proportion of variation in outcome variable explained by predictor variables, is commonly used for linear regression models (which have continuous outcome variables), its extension to logistic regression models (which have binary outcome variables) is not well-defined and is still an active research area in statistics. We therefore reported the predictive potential of the DNAm differences identified in this study, by including an additional sub-section on out-of-sample validation of methylation risk score (MRS) in an independent dataset in the Results section, in which we stated the prediction accuracy of the MRS alone and in combination with additional clinical factors. Please see details in (2)-(5) under “Revisions” in our reply to comment 1 above.

With regard to this reviewer’s question on sample sizes, please see below for Table 1, which tabulated sample sizes of recent DNAm studies in AD. To clarify, as we noted in the manuscript, in the analysis of DNA methylation (DNAm) changes in blood samples, power has been an issue for detecting significant DNAm differences after multiple comparison corrections. Several studies approached this problem using different strategies. For example, Vasanthakumar et al. (2020)¹ reported DNAm differences in ADNI longitudinal study, and $P < 10^{-5}$ was used as the significance threshold. More recently, Nabais et al. (2021)¹⁴ meta-analyzed DNAm data from cross-sectional studies by AIBL, ADNI, and AddNeuroMed consortiums that included AD cases, along with additional datasets in ALS and PD, and reported DNAm differences shared by different neurodegenerative disorders. Note that in Nabais et al. (2021), only one observation from each subject in ADNI was included. One difference between our analysis vs. those from Nabais et al.

(2021)¹⁴ is that we included all samples from the ADNI study and used a mixed models approach to analyze the longitudinal data. *Among all these studies, the blood samples meta-analysis we presented in this manuscript analyzed the largest number of samples for AD vs. CN comparison (total N = 1284 samples, with 427 DNAm samples for AD cases and 857 samples for cognitively normal controls, see Supp. Table 1).*

For the analysis of DNAm changes in brain samples, in our previous meta-analysis (Zhang et al. 2020 *Nat Comm* 11:6114), we identified 3751 FDR-significant CpGs and 110 DMRs by meta-analyzing 1030 prefrontal cortex (PFC) brain samples¹⁵. Another recent meta-analysis of brain samples (Smith et al. 2021 *Nat Comm* 12:3517) included three out of the four datasets that we had analyzed in the analysis of PFC samples. A difference between our study vs. Smith et al. (2021) is that our meta-analysis studied only prefrontal cortex samples, while Smith et al. (2021) also studied samples from other brain regions such as the temporal gyrus and entorhinal cortex. It has been observed that DNAm levels in different brain regions in the cortex are highly correlated¹⁶, and the PFC brain area has been repeatedly shown to have the largest number of differentially methylated CpGs^{17, 18}. Table 1 below also included sample sizes of individual DNAm studies of AD. *Among all these studies, our previous meta-analysis of brain samples (Zhang et al. 2020 *Nat Comm* 11:6114) also analyzed the largest number of prefrontal cortex brain samples (total n = 1030 samples).*

Table 1 Sample sizes of recent AD blood and brain samples studies.

study	sample size	major findings in AD	data accession
blood samples studies			
Vasanthakumar et al. (2020) Clinical Epigenetics 12:84	ADNI longitudinal study, (number of subjects at visit 1) AD: 94, CN: 220 (total number of samples) AD: 198, CN: 601	42 CpGs with $P < 10^{-5}$	adni.loni.usc.edu
Nabais et al. (2021) Genome Biology 22:90	(AIBL) AD: 161, CN: 471 (ADNI cross-sectional) AD: 33, CN: 202 (AddNeuroMed) AD: 84, CN: 89 total n (ALS, PD, AD) cases: 4442, CN: 3283	meta-analysis across datasets of all neurodegenerative disorders identified 12 CpGs at genome-wide significance ($P < 3.30 \times 10^{-7}$)	GSE153712
Roubroeks et al. (2020) Neurobiology of Aging 95:26-45	AddNeuroMed study AD: 86, CN: 89	No CpGs reached genome-wide significance (2.4×10^{-7}) 1 DMR at 5% Sidak adjusted P-value	GSE144858
Kobayashi et al. (2020) Scientific Reports 10:12217	CN: 200, AD: 151	replicated DNAm diff in the COASY gene	not available
Madrid et al. (2018) J. Alzheimer's Dis. 66(3): 927-934	CN: 39, AD: 45	at $P < 0.05$, 17 significant CpGs in both AD vs. CN comparison and comparison with AD biomarkers	not available
Mitsumori et al. (2020) PLoS One 15(9):e0239196	CN: 48, AD: 48	candidate gene study DNAm levels in three genes, CR1 , CLU , and PICALM , were significantly lower in AD subjects	not available
Brain samples studies			
Smith et al. (2021) Nat Comm 12:3517	PFC: London n = 113, Mt. Sinai n = 146, ROSMAP n = 711	meta-analysis of different brain regions, at genome-wide significance level ($P < 1.24 \times 10^{-7}$)	GSE59685 GSE105109 GSE80970

	STG: London1 n = 113, Mt. Sinai n = 146, Arizona1 n = 302, Arizona2 n = 88 EC: London1 n = 113, London2 n = 95	PFC: 236 sig. CpGs STG: 95 sig. CpGs EC: 10 sig. CpGs	GSE134379 GSE109627 GSE66351
De Jager et al. (2014) Nat Neurosci 17(9):1156-63	PFC: n = 708	71 CpGs replicated in a two-stage analysis. Genome-wide significance ($P < 1.20 \times 10^{-7}$) was used in first stage.	syn3157275
Lunnon et al. (2014) Nat Neurosci 17(9):1164-70	EC: n = 104, STG: n = 113, PFC: n = 110, CER: n = 108	identified CpGs at ANK1 with $P = 4.59 \times 10^{-7}$	GSE59685
Smith et al. (2018) Alzheimers Dement 14(12):1580-1588	PFC: n = 144, STG: n = 142	10 significant CpGs at $P < 2.2 \times 10^{-7}$, 78 CpGs at $P < 1 \times 10^{-5}$	GSE80970

Abbreviations: CN = cognitive normal, PFC = prefrontal cortex, EC = entorhinal cortex, STG = superior temporal gyrus, CER = cerebellum

Revision

In Results, under “Meta-analysis identified methylation differences significantly associated with AD at individual CpGs and genomic regions in the blood”, added statement about effect sizes of beta values on association with AD diagnosis

In the meta-analysis, the mean and median effect size of beta values (parameter estimate for methylation beta values in logistic regression model) across all CpGs were -0.33 and -0.27, respectively. In contrast, the mean and median effect size of beta values for the 50 AD-associated CpGs were -3.20 and -6.00, respectively.

4) I assume the red line in Figure 1 shows the cutoff of 10^{-7} ? Can the authors clarify?

In response, we added to the legend of Figure 1 that “The red line indicates significance threshold of 5% False Discovery Rate”.

Revision – updated legend for Figure 1

Figure 1 Miami plot for blood and brain samples meta-analyses. The X-axis shows chromosome numbers. The Y-axis shows $-\log_{10}(P\text{-value})$ of methylation-AD diagnosis associations in the blood (above X-axis) or methylation-AD neuropathology (Braak stage) associations in the brain (below X-axis). The genes corresponding to the top 20 most significant CpGs in blood or brain samples meta-analyses are highlighted. The P -values for the blood samples meta-analysis are from the current study, and the P -values for brain samples meta-analysis were obtained from Zhang et al. (2020) (PMID: 33257653). **The red line indicates a significance threshold of 5% False Discovery Rate.**

5) The authors tend to use FDR as a cutoff with the notably exception for their first analyses in blood, where they use 10^{-5} or even 0.05. It looks like this was done because ‘there was just no usable findings at 10^{-7} or -8 ’. The authors mentioned this in passing in the discussion, but could be even more transparent about this and include a statement of the strengths and limitations of this approach.

We thank the reviewer for this helpful suggestion. In response, we added clarifications on different significance thresholds used in the study as a limitation in the Discussion.

Revision – In Discussion, 2nd to the last paragraph

Third, in the meta-analysis of blood samples, only 5 CpGs reached 5% FDR, we therefore examined 45 additional CpGs at the less stringent significance threshold of P -value $< 10^{-5}$. Also, to select DNA methylation differences that are significant in the brain, blood, and cross-tissue meta-analyses (Figure 2), instead of using the FDR-significant CpGs in blood samples meta-analysis, we intersected nominally significant (i.e., P -value < 0.05) CpGs and DMRs in blood samples meta-analysis with FDR-significant DNA methylation differences in brain and cross-tissue meta-analyses. These more relaxed significance thresholds might correspond to higher false-positive rates. To help prioritize the most biologically relevant DNA methylation differences, we performed several integrative analyses of methylation with genetic variants, gene expressions, and transcription factor binding sites.

6) Why was the data pruned for >65y?

This reviewer raised an important point. To clarify, as we are mainly interested in late-onset AD, we excluded younger cases to avoid the possibility of including early-onset AD patients. We used the same criteria (age < 65 years) to also exclude control subjects, to avoid imbalance in age distribution for the two groups, and also the possibility that some of the controls might develop AD later on. This strategy is often implemented in the study of late-onset AD, see as an example, another recently published blood samples epigenome-wide association study (page 27 of Roubroeks et al. 2020, *Neurobiology of Aging* 95: 26-45).

7) I appreciate Figure 2 and encourage the authors to add more of those flow-chart figures to their manuscript to ease understanding. I could not find the supplementary figures.

In response to this reviewer's suggestion, we added a few more flow-chart figures: Supplementary Figures 2, 3 and 12, describe the workflow for cross-tissue analyses of DNA differences in blood samples with temporal gyrus and entorhinal cortex brain regions, and integrative analysis of DNA methylation and gene expression datasets generated from brain and blood samples.

(1) Supplementary Figure 2 – workflow of cross-tissue analysis of DNAm differences in blood samples with those in temporal gyrus brain samples

Supplementary Fig 2 Workflow for prioritizing differentially methylated CpGs associated with AD pathology (in temporal gyrus brain samples) and AD diagnosis (in blood samples). Genomic corrections were performed using the bacon method (PMID: 28129774) in the analysis of all individual datasets. The P -values for significant CpGs associated with AD Braak stage in the temporal gyrus region were obtained from Supplementary Table 3 of Smith et al. (2021) (PMID: 34112773).

(2) Supplementary Figure 3 - workflow of cross-tissue analysis of DNAm differences in blood samples with those in entorhinal cortex brain samples

Supplementary Fig 3 Workflow for prioritizing differentially methylated CpGs associated with AD pathology (in entorhinal cortex brain samples) and AD diagnosis (in blood samples). Genomic corrections were performed using the bacon method (PMID: 28129774) in the analysis of all individual datasets. The *P*-values for significant CpGs associated with AD Braak stage in the entorhinal cortex region were obtained from Supplementary Table 5 of Smith et al. (2021) (PMID: 34112773).

(3) Supplementary Figure 12 Workflow of integrative DNAm and gene expression data.

Supplementary Figure 12 Workflow of integrative DNA methylation (DNAm) and gene expression datasets. For blood samples analysis, we used the ADNI dataset with 265 matched DNAm and gene expression samples. *DNAm residuals* were computed by fitting a linear model with methylation *M*-value as the outcome, age, sex, estimated immune cell-type proportions, and batch effects as independent variables and extracting residuals. *RNA residuals* were computed similarly, except by using log-transformed gene expression values as the outcome. For brain samples analysis, we used the ROSMAP AD dataset with 529 matched DNAm and RNA-seq samples. DNAm residuals and RNA residuals were computed in the same way as for blood samples, except that we used estimated neuron proportions in the DNAm model and expression levels of marker genes for different brain cell types in the RNA model (i.e., *ENO2* for neurons, *GFAP* for astrocytes, *CD68* for microglia, *OLIG2* for oligodendrocytes, and *CD34* for endothelial cells). For each tissue, we next performed the following analysis steps: (1) first, we used principal component analysis to summarize DNAm residuals at the significant CpGs by the first PC (PC1), which is a weighted linear combination of the methylation residuals, these are the methylation PC scores; (2) Next, we tested the association between methylation PC scores (i.e., PC1) and genome-wide gene expressions residuals using linear models; (3) Finally, we ranked the genes by the absolute value of t-statistics for methylation PC scores, and performed pathway analysis using the fgsea software.

* In blood samples analysis, we computed PC1 of the 50 AD-associated CpGs with $P < 10^{-6}$. In brain samples analysis, we computed PC1 of the 3751 FDR significant CpGs in our previous brain samples meta-analysis (Zhang et al. 2020 PMID: 33257653).

8) I was confused by the fact that blood and brain methylation AD-related changes were opposite in direction yet the correlation between tissues were positive. How is this possible and what does this mean?

This reviewer made an important observation. We agree that “blood and brain methylation AD-related changes were opposite in direction yet the correlation between tissues were positive” is not reasonable. One possible contributing factor might be the relatively smaller sample size (n = 69 pairs) of the matched brain-blood samples, which might have resulted in inaccuracies. Also, brain-blood correlations in DNAm might also vary by AD severity. Future collections of matched brain-blood samples with larger number of samples at different AD stages are needed to prioritize the DNAm with changes in both brain and blood that we identified in this study. In response to this reviewer’s comment, we removed the section “Correlation of AD-associated CpGs and DMRs methylation levels in blood and brain samples” from the manuscript.

9) I was surprised by the lack of figures. I encourage the authors to evidence some of their findings with plots (which can often show the relevance of outliers or clustering). For example, SM Table 4a or 5 (and possibly many more) could be also visually presented.

We thank this reviewer for the good suggestion. In response, we added 8 figures to illustrate Supplementary Tables 4a, 5, and 6. The manuscript now includes 4 main figures and 12 Supplementary figures.

Revision

(1) For information in Supp Table 4a, added Supplementary Figure 4

Supplementary Figure 4 Among the 50 CpGs that reached $P < 10^{-5}$ and 9 DMRs with 5% Sidak corrected P-values in AD vs. CN blood samples meta-analysis, 3 CpG-gene or DMR-gene pairs reached 5% FDR significance in the analysis of ADNI blood samples with matched DNAm-RNA data. **A** cg07886485 – PXX, **B** cg23963071-SERPINB9, **C** chr1:205819345-205819464-PM20D1

(2) For information in Supp Table 5, added Supplementary Figures 5-7

Supplementary Figure 5 Results from MethReg analysis of ADNI blood samples showed in the cg16908123-RUNX3-C1orf100 triplet, DNAm attenuated TF activity (P-value for DNAm \times TF interaction = 6.58×10^{-6}). **A** Meta data for the CpG-TF-target gene triplet and results of fitting robust linear model to the triplet dataset. **B** When all samples are considered, no significant TF-target gene association was observed. **C** Comparison of target gene expression between the DNAm groups showed no association between DNAm and target gene expression. **D** In samples with low DNA methylation, target gene expression is observed to be repressed as TF activity increased. On the other hand, in samples with high DNA methylation, target gene expression is relatively independent of TF activity. Therefore, DNA methylation is predicted to attenuate TF activity on the target gene. **DNAm**: DNA methylation

Supplementary Figure 6 Results from MethReg analysis of ADNI blood samples dataset showed for the cg17643025-SREBF1-NNAT triplet, DNAm attenuated TF activity. **A** Meta data for the CpG-TF-target gene triplet and results of fitting robust linear model to the triplet dataset. **B** When all samples are considered, increased TF activity is associated with higher target gene expression levels. **C** Comparison of target gene expression between the DNAm groups showed no association between DNAm and target gene expression. **D** In samples with low methylation, target gene expression is observed to be increased as TF activity increased. In samples with high methylation, target gene expression is relatively independent of TF activity. Therefore, DNA methylation at cg17643025 is predicted to attenuate TF activity. **DNAm**: DNA methylation

Supplementary Figure 7 Results from MethReg analysis of ADNI blood samples dataset showed for the cg26312191-PAX5-SECTM1 triplet, DNAm attenuated TF activity. **A** Meta data for the CpG-TF-target gene triplet and results of fitting robust linear model to the triplet dataset. **B** When all samples are considered, increased TF activity is associated with higher target gene expression levels. **C** Comparison of target gene expression between the DNAm groups showed no association between DNAm and target gene expression. **D** In samples with low methylation, target gene expression is observed to be increased as TF activity increased. In samples with high methylation, target gene expression is relatively independent of TF activity. Therefore, DNA methylation at cg26312191 is predicted to attenuate TF activity. **DNAm**: DNA methylation

(3) For information in Supp Table 6, added Supplementary Figures 8-11

Supplementary Figure 8 Results from MethReg analysis of ROSMAP brain samples dataset showed for the cg01539849-MAFF-MADD triplet, DNAm enhanced TF activity. **A** Meta data for the CpG-TF-target gene triplet and results of fitting robust linear model to the triplet dataset. **B** When all samples are considered, increased TF activity is associated with lower target gene expression levels. **C** Comparison of target gene expression between the DNAm groups showed higher target gene expression in samples with lower DNAm. **D** In samples with low methylation, target gene expression is relatively independent of TF activity. In samples with high methylation, target gene expression is observed to be decreased as TF activity increased. Therefore, DNA methylation at cg01539849-MAFF-MADD is predicted to enhance TF activity. **DNAm**: DNA methylation

Supplementary Figure 9 Results from MethReg analysis of ROSMAP brain samples dataset showed for the cg22568423-PRAM1-SMAD1, DNAm attenuated TF activity. **A** Meta data for the CpG-TF-target gene triplet and results of fitting robust linear model to the triplet dataset. **B** When all samples are considered, TF activity is not associated with target gene expression levels. **C** Comparison of target gene expression between the DNAm groups showed lower target gene expressions in samples with lower DNAm. **D** In samples with low methylation, target gene expression is observed to be decreased as TF activity increased. In samples with high methylation, the TF-target gene association is weaker. Therefore, DNA methylation at cg22568423 is predicted to attenuate TF activity. **DNAm**: DNA methylation

Supplementary Figure 10 Results from MethReg analysis of ROSMAP brain samples dataset showed for the cg02459543-TMEM67-CEBPA triplet, DNAm enhanced TF activity. **A** Meta data for the CpG-TF-target gene triplet and results of fitting robust linear model to the triplet dataset. **B** When all samples are considered, TF activity is not associated with target gene expression levels. **C** Comparison of target gene expression between the DNAm groups showed lower target gene expressions in samples with lower DNAm. **D** In samples with high methylation, target gene expression is observed to be increased as TF activity increased. In samples with low methylation, TF-target gene expression is relatively weaker. Therefore, DNAm at cg02459543M is predicted to enhance TF activity. **DNAm**: DNA methylation

Supplementary Figure 11 Results from MethReg analysis of ROSMAP brain samples dataset showed for the cg22568423-MYO1F-CEBPA, DNAm attenuated TF activity. **A** Meta data for the CpG-TF-target gene triplet and results of fitting robust linear model to the triplet dataset. **B** When all samples are considered, increased TF activity is associated with lower target gene expression levels. **C** Comparison of target gene expression between the DNAm groups showed lower target gene expressions in samples with lower DNAm. **D** In samples with low methylation, target gene expression is observed to be decreased as TF activity increased. In samples with high methylation, target gene expression is relatively independent of TF activity. Therefore, DNA methylation at cg22568423 is predicted to attenuate TF activity. **DNAm**: DNA methylation

10) Please add the FDR column to SM Table 4b - blood, unless there's a reason why it's missing.

To clarify, because no DNAm-RNA pairs reached 5% FDR in ADNI blood samples, we therefore omitted the FDRs in Supplementary Table 4b (blood column) initially. In response, we added the FDRs back to give the readers more information on these results.

Revision – Supplementary Table 6(b)

Supplementary Table 6 (b) For the prioritized CpGs and DMRs that are significant in both brain and blood samples (Figure 2), 11 DNAm-RNA pairs reached 5% FDR in the analysis of ROSMAP brain samples with matched DNAm-RNA samples. No DNAm-RNA pairs reached 5% FDR in ADNI blood samples. Highlighted rows are CpGs and DMRs with the same direction of DNAm-RNA associations in the brain and blood.

DNAm		target gene			DNAm vs. RNA associations						
probeID	location	symbol	gene_ID	distance to TSS	in brain samples			in blood samples			Direction (brain, blood)
					estimate	P-value	fdr	estimate	P-value	fdr	
cg22568423	chr19:8590567	MYO1F	ENSG00000142347	51892	0.070	1.96E-06	1.64E-03	0.165	9.52E-03	3.60E-01	++
cg05157625	chr14:93153553	RIN3	ENSG00000100599	173434	0.078	5.74E-06	2.35E-03	0.184	2.04E-03	2.37E-01	++
cg05157625	chr14:93153553	SLC24A4	ENSG00000140090	364627	-0.082	2.61E-05	4.02E-03	0.043	4.97E-01	9.80E-01	-+
cg11744817	chr6:166876826	T	ENSG00000164458	-294637	0.002	1.48E-04	1.23E-02	0.021	7.03E-01	9.88E-01	++
cg24360871	chr7:27163929	HOXA7	ENSG00000122592	33624	0.002	2.18E-04	1.52E-02	0.003	9.64E-01	9.90E-01	++
cg00502254	chr1:12201600	TNFRSF1B	ENSG00000028137	-25458	-0.217	9.71E-04	3.69E-02	0.022	7.41E-01	9.88E-01	-+
cg22568423	chr19:8590567	PRAM1	ENSG00000133246	-22570	0.064	8.43E-04	3.69E-02	0.083	1.77E-01	8.13E-01	++
cg26643870	chr22:41845659	CSDC2	ENSG00000172346	-111106	-0.234	1.14E-03	4.08E-02	-0.023	7.46E-01	9.88E-01	--
cg26643870	chr22:41845659	L3MBTL2	ENSG00000100395	244449	-0.076	1.17E-03	4.08E-02	-0.030	6.52E-01	9.88E-01	--
cg10095954	chr17:3848324	P2RX1	ENSG00000108405	-28529	0.037	1.49E-03	4.99E-02	-0.029	6.26E-01	9.88E-01	+-
DMR	chr15:39871808-39872186	THBS1	ENSG00000137801	-1093	-0.096	2.09E-03	1.46E-02	-0.017	7.28E-01	9.13E-01	--

11) The results section could benefit from the odd short summary at the end of each sub-section. For example, it seems that the methylation-expression relationship was stronger or at least more significant in brain compared to blood tissue. The end of the mQTL subsection provides an excellent example for this.

In response to this reviewer’s suggestion, we included additional short summaries at the end of each sub-section.

Revisions

(1) In Results, under “Meta-analysis identified methylation differences significantly associated with AD at individual CpGs and genomic regions in blood”

Taken together, these results demonstrated the results of our meta-analysis are consistent with recent epigenomics literature in brain research. In addition to replicating methylation differences in genes previously implicated in AD (e.g., *PM20D1*, *EIF2D*), we also nominated additional genes that might be associated with AD.

(2) In Results, under “Cross-tissue meta-analysis prioritized AD-associated DNA methylation differences in both brain and blood”

These CpGs and DMRs that are significant in both brain and blood tissues highlighted AD-associated DNA methylation in the periphery that are also altered in the brain.

(3) In Results, under “Correlation of methylation levels of significant CpGs and DMRs in AD with expressions of nearby genes”

The greater number of methylation – gene expression associations detected in brain samples compared to blood samples could be due to the larger sample size of matched methylation-RNA brain samples available (529 ROSMAP brain samples vs. 265 ADNI blood samples).

In this section, we identified CpGs and DMRs that influence target gene expression directly. In the next section, we discuss identifying CpG methylations that influence target gene expression indirectly by modulating transcription factor activities.

(4) In Results, under “MethReg integrative analysis associated CpGs with putative transcription factors and target genes”

Our MethReg analysis revealed a number of TFs that might interact with AD-associated CpG methylations to jointly regulate target gene expressions (Supplementary Tables 7-8). Importantly, for these methylation-sensitive TFs, the TF-target associations are often only present in a subset of samples with high (or low) methylation levels, thus might be missed by analyses that use all samples (Supplementary Figures 5-11). Although many of the TFs have previously been implicated to AD, our integrative analysis provided additional information on the specific roles of the TFs in transcriptional regulation, and identified target genes for these TFs in AD, by nominating plausible TF-target gene associations that are mediated by DNA methylations.

(5) “Integrative analysis revealed gene expressions associated with DNA methylation differences in the blood and the brain converge in biological pathways”

These results suggested there might be a convergence in pathways across brain and blood in gene expression changes associated with the methylation PC scores.

12) mQTLs: the authors find more mQTLs in blood than in brain tissue. Is that a function of sample size (i.e. is GoDMC much larger than xQTL able to detect smaller effects)?

This reviewer is correct; we added clarifications about the different samples sizes for the brain and blood samples mQTL studies to the manuscript.

Revision – In Results, under “Correlation and overlap with genetic susceptibility loci”

The larger number of mQTLs detected in blood could be due to the larger sample size of GoDMC which used a meta-analysis design¹⁹, compared to xQTL which was computed based on a single cohort (ROSMAP)²⁰.

13) Discussion: the authors mention consistent findings with Nabais et al. However, I believe AIBL data featured in both analyses, in which case consistency in findings should not be too surprising.

We agree with this reviewer that both our analysis and those of Nabais et al. included the AIBL data, which may have contributed to the consistent findings. We wish to clarify in Nabais et al. study, only a small proportion (about 6%; 278 out of 4442 samples) of the cases are AD cases (see Figure 1 of Nabais et al.), the majority of the cases were ALS (n = 3032) and Parkinson’s disease (n = 1132). Therefore, findings in

Nabais et al. (2021) represent common susceptibility loci among different neurodegenerative diseases, while findings in our study are AD-specific. We added clarification for this point in the Discussion section.

Revision – in the Discussion section, the first paragraph

Among them, two CpGs (cg03546163 and cg14195992), mapped to the *FKBP5* and *SPIDR* genes, also reached genome-wide significance in another large meta-analysis of multiple EWAS (of Amyotrophic lateral sclerosis, Parkinson’s disease, and AD)¹⁴, suggesting these two CpGs corresponded to susceptibility loci common in neurodegenerative diseases.

14) *FKBP5* is >the< candidate gene for depression and childhood adversity. While the validity of candidate gene studies can be questioned, I feel that this deserved mentioning.

In response to this reviewer’s comment, we added additional text describing the role of *FKBP5* gene in depression and childhood adversity.

Revision – in the Discussion section, first paragraph

Moreover, genetic variants on *FKBP5* have also been implicated in stress-related disorders such as major depressive disorder^{21,22}, suicide behavior²², posttraumatic stress disorder²³, and childhood maltreatment²⁴. These results are consistent with the increased dementia risk observed in patients with depression²⁵.

15) Re causality: I suppose the authors could carry out a Mendelian Randomization analysis to assess if the identified methylation markers might indeed be causal.

In response to this reviewer’s comment, we performed Mendelian Randomization (MR) using the TwoSampleMR R package²⁶, which implemented several different algorithms²⁷⁻³⁰ that estimated the ratio between SNP effect on the outcome (i.e., AD status) and SNP effect on the exposure (i.e., DNAm), to infer mediation effect of CpG sites for genetic influences on AD status. Specifically, for SNP-AD associations, we used the results from the recent large-scale meta-analysis by Kunkle et al. (2019)³¹ (file “Kunkle_et_al_Stage1_results.txt” downloaded from <https://www.niagads.org/igap-rv-summary-stats-kunkle-p-value-data>); for SNP-DNAm associations, we used mQTL associations obtained from the GoDMC database³² for the AD-associated CpGs. However, we did not detect any evidence for mediation of the AD-associated CpGs using MR analysis; the lowest p-value from our analysis was 0.240; therefore, we did not include this result in the manuscript.

Our results are consistent with those reported by another recent study (Min et al. (2021) *Nature Genetics* 53, 1311–1321), which performed MR analyses for different phenotypic traits, including AD (“Alzheimer’s disease” is listed on row 108 as one of the traits studied in MR analysis in Supplementary Table 15 of Min et al. 2021), and also did not detect any DNAm sites mediating genetic associations with AD. Min et al. (2021) concluded that “these results indicate that those blood-measured DNAm sites that have shared genetic factors with traits cannot be typically thought of as mediating the genetic association with the trait. Instead, if DNAm is a co-regulatory phenomenon then the co-localizing signals between DNAm sites and complex traits may be due to a common cause, for example, genetic variants primarily acting on TF binding”) ¹⁹.

16) Methods: I was surprised to read that no age information was available in AIBL. I wonder what effect it has to control for epigenetically predicted age in an EWAS. Do the authors not risk losing potentially important epigenetic signals, especially since age itself is also strongly linked to the outcome?

This reviewer raised an important point. To clarify, we used a fairly accurate epigenetic clock developed by Zhang et al. (2019)³³, which was trained using a large number ($n = 12710$) of blood samples, and was shown to be more accurate than several alternative epigenetic clocks³³. To assess the impact of using chronological age or epigenetic age in regression analysis that associated DNAm with AD diagnosis, we compared the analysis results for two alternative models (see below) using the AddNeuroMed dataset, which had chronological age information available.

More specifically, we compared regression estimate for beta variable, representing the effect of methylation beta values on AD diagnosis, in the following two models:

Model 1:

```
glm(DIAGNOSIS ~ beta + age.pred.Elastic_Net + Sex + as.factor(PLATE) + B + NK + CD4T + CD8T + Mono + Neutro, data = dat.cn.ad, family = binomial)
```

where age.pred.Elastic_Net = predicted epigenetic age using the method of Zhang et al. (2019)

Model 2:

```
glm(DIAGNOSIS ~ beta + age + Sex + as.factor(PLATE) + B + NK + CD4T + CD8T + Mono + Neutro, data = dat.cn.ad, family = binomial)
```

where age = chronological age

```
> cor.test(df$Estimate_beta_real_age,df$Estimate_beta_predicted_age)

Pearson's product-moment correlation

data:  df$Estimate_beta_real_age and df$Estimate_beta_predicted_age
t = 2248.5, df = 392047, p-value < 2.2e-16
alternative hypothesis: true correlation is not equal to 0
95 percent confidence interval:
 0.9631189 0.9635695
sample estimates:
      cor
0.9633449
```

As the output above shows, the estimated effect for methylation beta values are highly correlated in model 1 and 2 (Pearson correlation = 0.963, P -value $< 2.2 \times 10^{-16}$), indicating including chronological age or predicted epigenetic age gave very similar results for the association between DNAm and AD diagnosis.

17) The uncorrected lambda for ADNI were very low. Can the authors comment on that?

To clarify, the low values of uncorrelated lambda for the ADNI dataset suggested the mixed-effects models P -values (computed based on a theoretical formula) in the analysis of longitudinal ADNI samples were conservative. Efron (2010) showed that in large-scale simultaneous testing situations (e.g., when many

CpGs are tested in an analysis), serious defects in the theoretical null distribution may become obvious, while empirical Bayes methods can provide much more realistic null distributions³⁴. To this end, we used the bacon method³⁵ to estimate empirical null distributions and to compute the bacon-corrected P -values, which provided a more accurate assessment of statistical significance. After the bacon correction, the estimated inflation factor was $\lambda = 0.97$ for the ADNI dataset.

In response to this reviewer's comment, we added more clarification about using empirical null distribution for statistical assessment in the Methods section.

Revision – In the Methods section, 4th paragraph under “Blood samples meta-analysis”

Next, for more accurate statistical assessment, genomic correction using the bacon method³⁵, as implemented in the bacon R package, was applied to obtain bacon-corrected effect sizes, standard errors, and P -values for each cohort. Efron (2010) showed that in large-scale simultaneous testing situations (e.g., when many CpGs are tested in an analysis), serious defects in the theoretical null distribution may become obvious, while empirical Bayes methods can provide much more realistic null distributions³⁴. By definition, the bacon-corrected test statistics have estimated an inflation factor of 1 because empirical null distributions were used in their estimation. Indeed, after bacon-correction, the estimated inflation factors were $\lambda = 0.97$ and 1.02, and $\lambda_{\text{bacon}} = 0.98$ and 0.97, for the ADNI and AIBL cohorts, respectively.

References

1. Vasanthakumar, A. et al. Harnessing peripheral DNA methylation differences in the Alzheimer's Disease Neuroimaging Initiative (ADNI) to reveal novel biomarkers of disease. *Clin Epigenetics* **12**, 84 (2020).
2. Ellis, K.A. et al. Enabling a multidisciplinary approach to the study of ageing and Alzheimer's disease: an update from the Australian Imaging Biomarkers and Lifestyle (AIBL) study. *Int Rev Psychiatry* **25**, 699-710 (2013).
3. Roubroeks, J.A.Y. et al. An epigenome-wide association study of Alzheimer's disease blood highlights robust DNA hypermethylation in the HOXB6 gene. *Neurobiol Aging* **95**, 26-45 (2020).
4. Huls, A. & Czamara, D. Methodological challenges in constructing DNA methylation risk scores. *Epigenetics* **15**, 1-11 (2020).
5. McCartney, D.L. et al. Epigenetic prediction of complex traits and death. *Genome Biol* **19**, 136 (2018).
6. Nabais, M.F. et al. Significant out-of-sample classification from methylation profile scoring for amyotrophic lateral sclerosis. *NPJ Genom Med* **5**, 10 (2020).
7. Sims, R., Hill, M. & Williams, J. The multiplex model of the genetics of Alzheimer's disease. *Nat Neurosci* **23**, 311-322 (2020).
8. Zheng, S.C., Breeze, C.E., Beck, S. & Teschendorff, A.E. Identification of differentially methylated cell types in epigenome-wide association studies. *Nat Methods* **15**, 1059-1066 (2018).
9. Mason, S.J. & Graham, N.E. Areas beneath the relative operating characteristics (ROC) and relative operating levels (ROL) curves: Statistical significance and interpretation. *Quarterly Journal of the Royal Meteorological Society* **128**, 2145-2166 (2002).
10. Kennedy, B.K. et al. Geroscience: linking aging to chronic disease. *Cell* **159**, 709-713 (2014).
11. Castrillo, J.I., Lista, S., Hampel, H. & Ritchie, C.W. Systems Biology Methods for Alzheimer's Disease Research Toward Molecular Signatures, Subtypes, and Stages and Precision Medicine: Application in Cohort Studies and Trials. *Methods Mol Biol* **1750**, 31-66 (2018).
12. Klein, H.U., Bennett, D.A. & De Jager, P.L. The epigenome in Alzheimer's disease: current state and approaches for a new path to gene discovery and understanding disease mechanism. *Acta Neuropathol* **132**, 503-514 (2016).
13. Chibnik, L.B. et al. Alzheimer's loci: epigenetic associations and interaction with genetic factors. *Ann Clin Transl Neurol* **2**, 636-647 (2015).
14. Nabais, M.F. et al. Meta-analysis of genome-wide DNA methylation identifies shared associations across neurodegenerative disorders. *Genome Biol* **22**, 90 (2021).
15. Zhang, L. et al. Epigenome-wide meta-analysis of DNA methylation differences in prefrontal cortex implicates the immune processes in Alzheimer's disease. *Nat Commun* **11**, 6114 (2020).
16. Hannon, E., Lunnon, K., Schalkwyk, L. & Mill, J. Interindividual methylomic variation across blood, cortex, and cerebellum: implications for epigenetic studies of neurological and neuropsychiatric phenotypes. *Epigenetics* **10**, 1024-1032 (2015).
17. Lunnon, K. et al. Methylomic profiling implicates cortical deregulation of ANK1 in Alzheimer's disease. *Nat Neurosci* **17**, 1164-1170 (2014).
18. Smith, R.G. et al. Elevated DNA methylation across a 48-kb region spanning the HOXA gene cluster is associated with Alzheimer's disease neuropathology. *Alzheimer's & dementia : the journal of the Alzheimer's Association* **14**, 1580-1588 (2018).
19. Min, J.L. et al. Genomic and phenotypic insights from an atlas of genetic effects on DNA methylation. *Nature genetics* **53**, 1311-1321 (2021).
20. Ng, B. et al. An xQTL map integrates the genetic architecture of the human brain's transcriptome and epigenome. *Nat Neurosci* **20**, 1418-1426 (2017).

21. Binder, E.B. et al. Polymorphisms in FKBP5 are associated with increased recurrence of depressive episodes and rapid response to antidepressant treatment. *Nature genetics* **36**, 1319-1325 (2004).
22. Hernandez-Diaz, Y. et al. Association between FKBP5 polymorphisms and depressive disorders or suicidal behavior: A systematic review and meta-analysis study. *Psychiatry Res* **271**, 658-668 (2019).
23. Binder, E.B. et al. Association of FKBP5 polymorphisms and childhood abuse with risk of posttraumatic stress disorder symptoms in adults. *JAMA* **299**, 1291-1305 (2008).
24. Appel, K. et al. Moderation of adult depression by a polymorphism in the FKBP5 gene and childhood physical abuse in the general population. *Neuropsychopharmacology* **36**, 1982-1991 (2011).
25. Holmquist, S., Nordstrom, A. & Nordstrom, P. The association of depression with subsequent dementia diagnosis: A Swedish nationwide cohort study from 1964 to 2016. *PLoS Med* **17**, e1003016 (2020).
26. Hemani, G. et al. The MR-Base platform supports systematic causal inference across the human phenome. *Elife* **7** (2018).
27. Bowden, J., Davey Smith, G. & Burgess, S. Mendelian randomization with invalid instruments: effect estimation and bias detection through Egger regression. *Int J Epidemiol* **44**, 512-525 (2015).
28. Bowden, J., Davey Smith, G., Haycock, P.C. & Burgess, S. Consistent Estimation in Mendelian Randomization with Some Invalid Instruments Using a Weighted Median Estimator. *Genet Epidemiol* **40**, 304-314 (2016).
29. Burgess, S., Dudbridge, F. & Thompson, S.G. Combining information on multiple instrumental variables in Mendelian randomization: comparison of allele score and summarized data methods. *Stat Med* **35**, 1880-1906 (2016).
30. Hartwig, F.P., Davey Smith, G. & Bowden, J. Robust inference in summary data Mendelian randomization via the zero modal pleiotropy assumption. *Int J Epidemiol* **46**, 1985-1998 (2017).
31. Kunkle, B.W. et al. Genetic meta-analysis of diagnosed Alzheimer's disease identifies new risk loci and implicates Abeta, tau, immunity and lipid processing. *Nature genetics* **51**, 414-430 (2019).
32. Min, J.L., Hemani, G., ..., Mill, J. & Relton, C.L. Genomic and phenomic insights from an atlas of genetic effects on DNA methylation. *medRxiv*, <https://doi.org/10.1101/2020.1109.1101.20180406> (2021).
33. Zhang, Q. et al. Improved precision of epigenetic clock estimates across tissues and its implication for biological ageing. *Genome Med* **11**, 54 (2019).
34. Efron, B. Correlated z-values and the accuracy of large-scale statistical estimates. *J Am Stat Assoc* **105**, 1042-1055 (2010).
35. van Iterson, M., van Zwet, E.W., Consortium, B. & Heijmans, B.T. Controlling bias and inflation in epigenome- and transcriptome-wide association studies using the empirical null distribution. *Genome Biol* **18**, 19 (2017).

REVIEWER COMMENTS

Reviewer #2 (Remarks to the Author):

The goal of this study was to identify blood-based DNA methylation markers that also change with underlying neuropathology in the brain and biomarkers for AD.

To do this, the authors profiled DNA methylation using the same Infinium MethylationEPIC BeadChip. They performed a cross-tissue meta-analysis by combining 1284 blood DNA methylation from two independent cohorts, generated by the ADNI and AIBL studies, and 1030 prefrontal cortex DNA methylation from four additional datasets.

One of the main findings is that expressions levels of 13 genes and 10 pathways were significantly associated with the AD-associated methylation differences in both brain and blood. A number of these genes and pathways are involved in the immune responses in AD. Moreover, DNA methylation at cg05157625, located in RIN3 gene, is significantly associated with both AD diagnosis and neuropathology, and expression of the target gene in both tissues. The authors suggest cg05157625 could be a candidate biomarker for AD.

The overall approach is well performed, it meets the expected standards in the field, and it is novel as there are not that many studies in human samples studying this topic. The main findings support what is known, because they replicate methylation differences in genes previously implicated in AD, but also nominate additional genes that might be associated with AD. The approach is novel, performing integrative analysis of DNA methylation, gene expression, and TF (revealing that some TFs that might interact with AD-associated CpG methylations to jointly regulate target gene expressions).

We thank this reviewer for the enthusiasm and encouragement for our study. We have made substantial changes in response to this reviewer's helpful comments, which we will discuss in detail below.

One of the main problems is the important overlap between aging and AD-associated methylation differences.

This reviewer raised an important point. To clarify, as advanced age is the strongest risk factor for AD, the molecular processes in aging and AD are intertwined. In response to this reviewer's comment, we studied the recent literature on the inter-relationship between aging and AD. We found that several papers in the literature postulated that some age-associated DNA methylation changes are accentuated by AD¹⁻³. The Age-by-Disease Interaction hypothesis³ postulates that in many neuropsychiatric and neurodegenerative diseases, the trajectory of age-associated molecular phenotypes (e.g., gene expression) are modified and pushed in the disease-promoting direction, and that DNAm is a potential mechanism contributing to the age-associated gene expression changes in multiple diseases, including AD². *Therefore, aging is a driving force rather than a confounding factor in the pathogenesis of AD.* Supporting this hypothesis, brain samples with AD neuropathology were shown to have older estimated epigenetic ages than their chronological ages⁴, suggesting an acceleration of DNAm changes that normally occur in aging. Also, among CpGs that make up the epigenetic clocks, many are located near genes involved in neurodegeneration⁵. We added more clarifications on this point in the manuscript. Please see details under "Revision" below.

Revision

(1) In the Discussion section, 6th paragraph

Our results are also consistent with previous studies in aging, the strongest risk factor for AD. DNA methylation changes throughout the lifetime, and it has been observed that as people age, methylation decreases at intergenic regions but increases at many promoter-associated CpGs islands regions⁶. For example, a comparison of DNA methylation in CD4+ T cells of a centenarian with a newborn revealed pervasive hypomethylation across the genome⁷ in the aged individual. Similarly, Reynolds et al. (2014)⁸ studied age-related CpGs that are associated with gene expression (age-eMS) in human monocytes and T cells and found age-eMS tended to be hypomethylated as people become older. These previous observations of hypomethylation during aging, combined with our observation of hypomethylation in AD subjects compared to cognitively normal subjects with similar ages (Supplementary Table 1), are consistent with the hypothesis that some age-associated DNA methylation changes are accentuated by AD¹⁻³.

Specific comments:

1) The authors should include much more detail about the included samples. What is the diagnosis for those brains? Please include all the other variables relevant for this kind of sample: PMI, disease duration, Braak scores, CERAD, CDR, among others. I would suggest including other adjusting variables such as CDR, braak stage...

We thank this reviewer for the good suggestion. In response, we added clarifications about the sources of the four brain samples datasets in the text. We also included detailed information of the brain samples in Supplementary Table 3. For the ROSMAP dataset (Synapse: syn3157275), information was available for PMI, Braak scores, CERAD, and clinical diagnosis at the time of death. For the other three public datasets (London GSE59685; Mt Sinai GSE80970; and Gasparoni GSE66351), only Braak stage and clinical diagnosis information were available. Information about disease duration was not available for any of the datasets.

We also agree with this reviewer about adjusting for the Braak stage, and we did accordingly in our integrative analysis. Please see details in (3) below.

Clarifications / Revisions

(1) In the Results section, the first paragraph under “Cross-tissue meta-analysis prioritized AD-associated DNA methylation differences in both brain and blood”

To identify CpGs and genomic regions with DNA methylation differences in both brain and blood samples, we next performed a cross-tissue meta-analysis of six datasets by additionally including four brain prefrontal cortex (PFC) samples datasets, generated by the ROSMAP⁹, Mt. Sinai¹⁰, London¹¹, and Gasparoni¹² methylation studies, along with the two blood samples datasets described above. We previously meta-analyzed these four PFC brain datasets and identified a number of CpGs significantly associated with AD Braak stage¹³, a standardized measure of neurofibrillary tangle burden determined at autopsy¹⁴. Supplementary Table 3 includes detailed information (e.g., Braak stage, clinical diagnosis, PMI) for the brain samples.

(2) Added Supplementary Table 3 - Information on brain samples used in cross-tissue meta-analysis

Braak stage	brain samples dataset sample sizes			
	ROSMAP	London	Mt Sinai	Gasparoni
0-2	145	27	56	20
3-4	420	18	43	8
5-6	161	62	42	28
CERAD				
1 (definite)	221	-	-	-
2 (probable)	240	-	-	-
3 (possible)	75	-	-	-
4 (no AD)	190	-	-	-
Clinical Diagnosis				
Control	229	24	67	32
MCI	171	-	-	-
AD	307	53	74	24
Other	19	30	-	-
PMI mean(std)	7.50 (5.81)	-	-	-

(3) In Methods, the last paragraph in “Correlations between methylation levels of significant CpGs and DMRs in AD with expressions of nearby genes”

We then tested for association between methylation residuals and gene expression residuals, adjusting for the Braak stage using a separate robust linear model $residuals_{expression} \sim residuals_{DNAm} + Braak\ stage$. The analysis of DMRs was performed similarly, except by replacing CpG methylation levels with the median methylation level of all CpGs located within the DMR.

2) Analyzing the predictive value of these findings in an independent cohort would improve the manuscript.

In response to this reviewer’s suggestion, we added a new section on out-of-sample validations in Results, which evaluated the predictive value of the identified methylation associations in an independent cohort (the AddNeuroMed cohort).

Revisions

(1) In Results, added a new section on out-of-sample validation

Out-of-sample validations of AD-associated DNA methylation differences in an external cohort

To evaluate the feasibility of the identified methylation differences for predicting AD diagnosis, we next performed out-of-sample validations using an external DNA methylation dataset generated by the AddNeuroMed study, which included 83 AD cases and 88 control samples with ages greater than 65 years¹⁵. To this end, we computed the methylation risk scores (MRS)¹⁶, which were shown to have excellent discrimination of smoking status, and moderate discrimination of obesity, alcohol consumption, HDL cholesterol¹⁷, and amyotrophic lateral sclerosis case-control status¹⁸ recently.

More specifically, for each sample in the AddNeuroMed dataset, we computed MRS by summing the methylation beta values of the prioritized CpGs weighted by their estimated effect sizes in the AIBL dataset. Several logistic regression models were estimated using the AIBL dataset and then tested on the

AddNeuroMed dataset. We considered logistic regression models with three sources of variations that might affect prediction for AD diagnosis: known clinical factors (i.e., age and sex), estimated cell-type proportions for each sample, and MRS. When tested individually, the models that included age and sex, cell types, or MRS alone had AUCs of 0.649, 0.576, and 0.614, respectively (Figure 3). When combined with estimated cell types or MRS, prediction performance for the model with clinical factors (i.e., age and sex) improved to AUCs of 0.663 and 0.688, respectively. Notably, MRS was computed based on fewer than one hundred CpGs, while the six variables corresponding to cell type proportions were estimated using all CpGs on the array. The best performing model included age, sex, MRS, and cell types (AUC = 0.696, 95% CI: 0.616 - 0.770) (Figures 3-4), significantly more predictive than a random classifier with an AUC of 0.5 (P -value = 2.78×10^{-5}).

The same logistic regression model trained using both ADNI and AIBL datasets (instead of AIBL alone) performed slightly worse with an AUC of 0.678, which might be due to batch effect due to different training datasets. The model that included MRS based on AD-associated CpGs (instead of prioritized CpGs) also performed worse with an AUC of 0.609, probably because CpGs with cross-tissue differences also leveraged information from additional brain samples datasets. In addition to MRS, we also evaluated the performance of MPS (methylation PC scores) described above, which sums methylation beta values in the testing dataset weighed by loadings of the first principal component in PCA analysis. The best performing logistic regression model involving MPS was also estimated using the AIBL dataset, included variables age, sex, cell types, and MPS computed based on the same 91 prioritized cross-tissue CpGs, and achieved an AUC of 0.662.

(2) Added Figure 3 and Figure 4 for results of out-of-sample validation

Figure 3 Performance of different logistic regression models for predicting AD diagnosis in out-of-sample validation. The training samples included 135 AD cases and 356 control samples from the AIBL dataset and the testing samples included 83 AD cases and 88 control samples from the AddNeuroMed dataset. MRS was computed as the sum of methylation beta values for 91 prioritized CpGs from cross-tissue analysis weighted by their estimated effect sizes in the AIBL dataset. Abbreviation: AUC = Area Under ROC curve

Figure 4 Receiver Operating Characteristic curves (ROCs) for logistic regression models predicting AD diagnosis in out-of-sample validation using the AddNeuroMed dataset (83 AD cases, 88 controls). The training dataset included 135 AD cases and 356 control samples from the AIBL dataset. The best performing logistic regression model (AUC = 0.696) included methylation risk score (MRS), age, sex, and estimated cell-type proportions, where MRS was computed as the sum of methylation beta values for prioritized CpGs weighted by their estimated effect sizes in the AIBL dataset.

(3) In Discussion, 2nd to the last paragraph, added text to discuss out-of-sample validation

This study has several limitations.... Fifth, the MRS-based risk prediction model could also be further improved. Because DNA methylation samples in the testing dataset (AddNeuroMed) were measured by 450k arrays, which are different from the EPIC arrays used by the AIBL study, we only included CpGs that mapped to both arrays in the computation of our MRS. The performance of our MRS-based prediction

models can be assessed more accurately using future testing dataset measured by the EPIC arrays. Also, in the out-of-sample validation analysis, we did not include other important factors such as APOE genotype, which might also significantly predict AD diagnosis¹⁹ because we did not have access to APOE information in the AIBL and AddNeuroMed datasets. Our internal validation using the ADNI dataset suggested additionally including APOE ($\epsilon 4$ allele) into our best performing logistic regression model, which included MRS, age, sex, and estimated cell types (Figures 3-4), might substantially improve prediction performance. More specifically, a 10-fold cross-validation using the ADNI dataset showed the estimated average AUCs for the best performing logistic regression models with and without APOE status were 0.691 and 0.810, respectively (Supplementary Table 16). Finally, the associations we identified do not necessarily reflect causal relationships. Additional studies are needed to establish the causality of the nominated DNA methylation markers.

(4) In Methods, last paragraph

Out-of-sample validation

The best-performing risk prediction model was trained using samples from the AIBL dataset and tested on the AddNeuroMed dataset. More specifically, first, we computed the Methylation Risk Scores (MRS) as the sum of methylation beta values (for prioritized CpGs in Supplementary Table 4) weighted by their estimated effect sizes (i.e., parameter estimate for methylation beta values in logistic regression after bacon-correction) in the AIBL dataset¹⁶. We included 91 prioritized CpGs that were available in the AddNeuroMed dataset from GEO. Next, the logistic regression model $\text{logit}(\text{Pr}(\text{AD})) \sim \text{MRS} + \text{age} + \text{sex} + \text{B} + \text{NK} + \text{CD4T} + \text{CD8T} + \text{Mono} + \text{Neutro}$ was fitted to the AIBL dataset using `glm()` function, and `predict.glm()` was used to apply the logistic regression model to AddNeuroMed dataset. The last six variables in the logistic regression model correspond to estimated proportions of different blood cell types of B-cells, Natural Killer (NK) cells, CD4+ T-cells, CD8+ T-cells, Monocytes, and Neutrophils, obtained using the EpiDish R package²⁰. The R package pROC was used to estimate receiver operating characteristic curves (ROCs) and area under the ROC curves (AUCs). Similarly, logistic regression models with a subset of the variables in the above model (e.g., only age and sex) were similarly developed using the AIBL dataset and tested on the AddNeuroMed dataset. To determine if a logistic regression model predicted AD diagnosis significantly better than chance, we used the Wilcoxon rank-sum test to compare estimated probabilities for AD cases versus controls²¹. To assess the added prediction accuracy of APOE gene, we performed internal validations (i.e., 10-fold cross-validations) that compared our best performing model $\text{logit}(\text{Pr}(\text{AD})) \sim \text{MRS} + \text{age} + \text{sex} + \text{B} + \text{NK} + \text{CD4T} + \text{CD8T} + \text{Mono} + \text{Neutro}$ with the model that additionally included APOE ($\epsilon 4$ allele genotype) using the ADNI dataset. To obtain an independent set of samples, only the last visit of each subject in the ADNI dataset was used for this analysis. The function `createFolds()` in `caret` R

package was used to divide the data into 10 folds. Average AUCs over the 10 iterations in the 10-fold cross-validations for the models with and without APOE were then estimated and compared.

3) It would have been interesting to integrate these results with different brain areas, not only prefrontal cortex.

In response to this reviewer's comments, we also performed a cross-tissue analysis of the blood samples datasets with DNAm differences previously identified in temporal gyrus and entorhinal cortex brain regions (Supplementary Table 3 and 5 of Smith et al. (2021)²²).

Revision

(1) In the Results section, under "Cross-tissue meta-analysis identified AD-associated DNA methylation differences in both brain and blood"

Beyond PFC, additional brain regions in the cortex such as temporal gyrus (TG) and entorhinal cortex (EC) are often also affected by neurodegeneration in AD. Smith et al. (2021) studied differential methylation in the cortex associated with AD Braak stage and identified 236, 95, and 10 significant CpGs (at 5% Bonferroni adjusted *P*-value) in the PFC, TG and EC, respectively²². Our cross-tissue analyses of the blood samples with brain regions in the TC and EC nominated significant DNA methylation differences at 8 CpGs and 1 CpG, respectively (Supplementary Figures 2-3, Supplementary Table 5). These CpGs and DMRs that are significant in both brain and blood tissues highlighted AD-associated DNA methylation in the periphery that are also altered in the brain.

(2) Added Supplementary Figure 2 – workflow of cross-tissue analysis of DNAm differences in blood samples with those in temporal gyrus brain samples

Supplementary Fig 2 Workflow for prioritizing differentially methylated CpGs associated with AD pathology (in temporal gyrus brain samples) and AD diagnosis (in blood samples). Genomic corrections were performed using the bacon method (PMID: 28129774) in the analysis of all individual datasets. The *P*-values for significant CpGs associated with AD Braak stage in the temporal gyrus region were obtained from Supplementary Table 3 of Smith et al. (2021) (PMID: 34112773).

(3) Added Supplementary Figure 3 - workflow of cross-tissue analysis of DNAm differences in blood samples with those in entorhinal cortex brain samples

Supplementary Fig 3 Workflow for prioritizing differentially methylated CpGs associated with AD pathology (in entorhinal cortex brain samples) and AD diagnosis (in blood samples). Genomic corrections were performed using the bacon method (PMID: 28129774) in the analysis of all individual datasets. The *P*-values for significant CpGs associated with AD Braak stage in the entorhinal cortex region were obtained from Supplementary Table 5 of Smith et al. (2021) (PMID: 34112773).

(4) Added Supplementary Table 5 a-b for analyses results

Supplementary Table 5 A total of 8 CpGs and 1 CpG were significant in cross-tissue meta-analyses of DNAm differences in blood samples with those from the temporal gyrus (TG) and entorhinal cortex brain regions, respectively. The brain samples results were obtained from Supplementary Table 3 and 5 in Smith et al. (2021) (PMID: 34112773). Stouffer's method was used to combine weighted z-scores (transformed from *P*-values) in all six datasets, where the weights were specified based on the square root of the total number of subjects in each dataset.

(a) temporal gyrus																	
CpG	chr	pos	meta-analysis P-values			Bonferroni p.adjust	P-values in brain datasets				P-values in blood datasets			Annotation			
			brain	blood	cross tissue		London_1	Mount_Sinai	Arizona_1	Arizona_2	direction	ADNI	AIBL	direction	GREAT	Islands.UC	UCSC_RefGene_Group
cg25840926	chr2	20647987	2.54E-08	5.44E-04	1.17E-08	9.85E-07	2.72E-03	1.43E-03	6.60E-03	2.63E-02	+++	1.95E-02	1.07E-02	--	RHOB (+1153);HS1BP3 (+20 Island	RHOB;RHOB	1stExon;3'UTR
cg07207652	chr17	45798257	1.28E-09	4.07E-03	3.49E-08	2.93E-06	4.92E-03	2.38E-02	7.45E-06	1.09E-01	+++	4.58E-02	3.69E-02	--	TBX21 (-12353);TBKBP1 (+2 OpenSea		
cg08433504	chr15	39872071	6.50E-09	3.78E-03	1.91E-07	1.61E-05	5.60E-04	1.42E-01	1.84E-05	1.43E-01	+++	2.74E-01	4.50E-03	--	THBS1 (-1223)	N_Shore	THBS1 TSS1500
cg01105418	chr1	244214593	3.54E-11	4.12E-02	2.60E-07	2.18E-05	1.08E-02	7.41E-03	2.34E-06	6.71E-03	+++	3.82E-02	4.13E-01	++	ZBTB18 (+9)	S_Shore	ZNF238;ZNF2:1stExon;5'UTR
cg01419713	chr8	42038135	1.38E-10	3.56E-02	8.64E-07	7.26E-05	1.17E-01	3.00E-02	4.34E-07	1.83E-03	+++	2.01E-02	5.26E-01	--	PLAT (+27107);AP3M2 (+27 OpenSea	PLAT;PLAT	Body;Body
cg17104258	chr1	167090646	2.60E-08	3.19E-02	3.80E-06	3.19E-04	6.27E-03	1.47E-02	5.19E-04	2.54E-02	----	1.06E-01	1.55E-01	--	POU2F1 (-99420);DUSP27 (Island	DUSP27	Body
cg26360402	chr14	105973994	6.54E-08	3.17E-02	9.69E-06	8.14E-04	5.87E-04	3.37E-03	3.92E-03	2.08E-01	----	4.22E-02	2.93E-01	++	TMEM121 (-18946);C14orf6 OpenSea		
cg02798280	chr19	39087135	2.95E-08	1.64E-02	1.06E-05	8.94E-04	1.56E-05	1.55E-02	4.68E-04	9.72E-01	+++	1.07E-01	7.42E-02	--	MAP4K1 (+21429);RYR1 (+1 Island	MAP4K1;MAP	Body;Body

(b) entorhinal cortex																
CpG	chr	pos	meta-analysis P-values			Bonferroni p.adjust	P-values in brain datasets			P-values in blood datasets			Annotation			
			brain	blood	cross tissue		London_1	London_2	direction	ADNI	AIBL	direction	GREAT	Islands.UC	UCSC_RefGene_Name	RefGene_Group
cg04523589	chr3	48265146	3.26E-08	0.001838	2.84E-06	2.27E-05	4.61E-05	1.71E-04	+++	1.77E-03	2.40E-01	++	CAMP (+310)	OpenSea	CAMP	1stExon

4) How do you define "triplets"?

To clarify, to create the CpG-TF-target gene triplets, we first identified TFs that bind near the CpGs, by using information from the ReMap2020 database²³, which contains regulatory regions for 1135 transcriptional regulators obtained using genome-wide DNA-binding experiments such as ChIP-seq. Next, we linked a CpG to a gene if the CpG was within its promoter region; otherwise, we considered the CpG to be in the distal regions (> 2k bp from any promoter regions) and linked it to 5 genes upstream and 5 genes downstream of the CpG location. The CpG-TF pairs are then combined with CpG-target gene pairs to create triplets of CpG-TF-target genes. In response, we added clarification on how triplets are defined.

Revision

In the Methods section, next to “MethReg integrative analysis”

MethReg integrative analysis To create the CpG-TF-target gene triplets, we first linked a given CpG to transcription factors (TFs) with binding sites within ± 250 bp of the CpG, by using information from the ReMap2020 database²³, which contains regulatory regions for over one thousand transcriptional regulators obtained using genome-wide DNA-binding experiments such as ChIP-seq. Next, we linked a CpG to a gene if the CpG was within its promoter region; otherwise, we considered the CpG to be in the distal regions (> 2 k bp from any promoter regions) and linked it to 5 genes upstream and 5 genes downstream of the CpG location. **The CpG-TF pairs are then combined with CpG-target gene pairs to create triplets of CpG-TF-target genes.** We used the same 265 and 529 matched methylation-RNA samples from ADNI and ROSMAP studies described in the section above for blood and brain samples analysis. Methylation residuals and gene expression residuals were obtained in the same way as described above and used as input for the MethReg analysis. TF activities were estimated using the GSVA²⁴ R package. The MethReg analyses²⁵ were performed using the MethReg R package.

References

1. Kennedy, B.K. *et al.* Geroscience: linking aging to chronic disease. *Cell* **159**, 709-13 (2014).
2. McKinney, B.C. *et al.* DNA methylation in the human frontal cortex reveals a putative mechanism for age-by-disease interactions. *Transl Psychiatry* **9**, 39 (2019).
3. McKinney, B.C. & Sibille, E. The age-by-disease interaction hypothesis of late-life depression. *Am J Geriatr Psychiatry* **21**, 418-32 (2013).
4. Levine, M.E., Lu, A.T., Bennett, D.A. & Horvath, S. Epigenetic age of the pre-frontal cortex is associated with neuritic plaques, amyloid load, and Alzheimer's disease related cognitive functioning. *Aging (Albany NY)* **7**, 1198-211 (2015).
5. Hannum, G. *et al.* Genome-wide methylation profiles reveal quantitative views of human aging rates. *Mol Cell* **49**, 359-367 (2013).
6. Jones, M.J., Goodman, S.J. & Kobor, M.S. DNA methylation and healthy human aging. *Aging Cell* **14**, 924-32 (2015).
7. Heyn, H. *et al.* Distinct DNA methylomes of newborns and centenarians. *Proc Natl Acad Sci U S A* **109**, 10522-7 (2012).
8. Reynolds, L.M. *et al.* Age-related variations in the methylome associated with gene expression in human monocytes and T cells. *Nat Commun* **5**, 5366 (2014).
9. De Jager, P.L. *et al.* Alzheimer's disease: early alterations in brain DNA methylation at ANK1, BIN1, RHBDF2 and other loci. *Nat Neurosci* **17**, 1156-63 (2014).
10. Smith, R.G. *et al.* Elevated DNA methylation across a 48-kb region spanning the HOXA gene cluster is associated with Alzheimer's disease neuropathology. *Alzheimers Dement* **14**, 1580-1588 (2018).
11. Lunnon, K. *et al.* Methylomic profiling implicates cortical deregulation of ANK1 in Alzheimer's disease. *Nat Neurosci* **17**, 1164-70 (2014).
12. Gasparoni, G. *et al.* DNA methylation analysis on purified neurons and glia dissects age and Alzheimer's disease-specific changes in the human cortex. *Epigenetics Chromatin* **11**, 41 (2018).
13. Zhang, L. *et al.* Epigenome-wide meta-analysis of DNA methylation differences in prefrontal cortex implicates the immune processes in Alzheimer's disease. *Nat Commun* **11**, 6114 (2020).
14. Braak, H. & Braak, E. Staging of Alzheimer's disease-related neurofibrillary changes. *Neurobiol Aging* **16**, 271-8; discussion 278-84 (1995).
15. Roubroeks, J.A.Y. *et al.* An epigenome-wide association study of Alzheimer's disease blood highlights robust DNA hypermethylation in the HOXB6 gene. *Neurobiol Aging* **95**, 26-45 (2020).
16. Huls, A. & Czamara, D. Methodological challenges in constructing DNA methylation risk scores. *Epigenetics* **15**, 1-11 (2020).
17. McCartney, D.L. *et al.* Epigenetic prediction of complex traits and death. *Genome Biol* **19**, 136 (2018).
18. Nabais, M.F. *et al.* Significant out-of-sample classification from methylation profile scoring for amyotrophic lateral sclerosis. *NPJ Genom Med* **5**, 10 (2020).
19. Sims, R., Hill, M. & Williams, J. The multiplex model of the genetics of Alzheimer's disease. *Nat Neurosci* **23**, 311-322 (2020).
20. Zheng, S.C., Breeze, C.E., Beck, S. & Teschendorff, A.E. Identification of differentially methylated cell types in epigenome-wide association studies. *Nat Methods* **15**, 1059-1066 (2018).
21. Mason, S.J. & Graham, N.E. Areas beneath the relative operating characteristics (ROC) and relative operating levels (ROL) curves: Statistical significance and interpretation. *Quarterly Journal of the Royal Meteorological Society* **128**, 2145-2166 (2002).
22. Smith, R.G. *et al.* A meta-analysis of epigenome-wide association studies in Alzheimer's disease highlights novel differentially methylated loci across cortex. *Nat Commun* **12**, 3517 (2021).

23. Cheneby, J. *et al.* ReMap 2020: a database of regulatory regions from an integrative analysis of Human and Arabidopsis DNA-binding sequencing experiments. *Nucleic Acids Res* **48**, D180-D188 (2020).
24. Hanzelmann, S., Castelo, R. & Guinney, J. GSEA: gene set variation analysis for microarray and RNA-seq data. *BMC Bioinformatics* **14**, 7 (2013).
25. Silva, T.C., Young, J.I., Martin, E.R., Chen, X. & Wang, L. MethReg: estimating the regulatory potential of DNA methylation in gene transcription. *Nucleic Acids Research*, accepted for publication (2022).

Reviewer #3 (Remarks to the Author):

The authors perform a comprehensive investigation of AD associated DNA methylation changes. First leveraging data by meta analysis in two large consortia for AD in blood, and then performing cross tissue analysis to assess relevance of the identified associations. Their approach yielded interesting results that match with the understanding of underlying AD etiology and add weight to previously identified genetic associations, while also supplying new genes of interest. mQTL analyses support cross tissue findings and identify interesting potential biomarkers. In general, this work is quite comprehensive and adds to the epigenetics and AD literature. There is very little the authors have not done to ensure robustness of their work and to assess that identified changes have functional or tissue specific relevance in the brain. Their findings match with and expand upon what is known about AD. Their discussion is well written and mentions the appropriate limitations inherent in methylation studies. I think this is a very solid piece of work and recommend publication.

We appreciate this reviewer's enthusiasm and encouragement for our study. Thank you very much! We discuss in detail below some changes we made in response to this reviewer's helpful comments.

Very minor comment:

You may wish to change the font from what appear to be pastes from the GO output tables into the text. This is not a reflection on the science or paper preparation in anyway, just something I noticed.

As this reviewer suggested, we updated the font used for pathway names so that they are consistent with the rest of the text.

Revision – In the Results section, 3rd paragraph under “Integrative analysis revealed gene expressions associated with DNA methylation differences in the blood and the brain converge in biological pathways”

Among the pathways that reached 5% FDR significance in brain or blood analyses, a number of pathways were involved in inflammatory responses in AD, such as **neutrophil degranulation, antigen processing and presentation, interferon signaling, and activation of nuclear factor kappa B pathways**. Additional significant biological processes included biological processes previously shown to be important in AD such as **glycolysis, antiviral mechanism, endocytosis, mRNA translation^{1,2}, and retrograde transport³**.

Minor comment:

The biomarker analysis identifying mQTL associated RIN3 with cross tissue methylation leads the authors to suggest it may be a good biomarker for AD. This suggestion is acknowledged as likely beyond scope for the current paper, which is packed with interesting analysis, but to further investigate and support this biomarker assertion, it would be interesting to generate a model of RIN3 methylation for AD status and apply it to peripheral tissue samples in an independent cohort to assess its potential predictive efficacy through ROC and AUC metrics, etc. Of course, this would be most interesting in a high risk AD sample that had not yet developed the pathology and the availability of such samples is unknown to this reviewer. Still, such a biomarker would certainly be interesting, as current AD risk assessment at this time in my understanding just relies on APOE genetic assessment, scales, or potentially imaging and new biomarkers are needed to understand if novel prophylactic approaches like

MAB therapy, for example, could be used in those at future risk.

This reviewer brought up an important point. In response, we evaluated the feasibility of the CpG cg05157625 on RIN3 for predicting AD diagnosis in an external dataset generated by the AddNeuroMed study (Roubroeks et al. 2020, *Neurobiology of Aging* 95: 26-45, GEO accession: GSE144858). Our results showed the AUC for a logistic regression model with beta value at cg05157625 as the predictor variable is 0.497, similar to those obtained from a random classifier.

To further investigate the predictive values of the DNAm differences identified in this study, we next performed an out-of-sample validation study using the AddNeuroMed dataset for methylation risk scores (MRS)⁴, which are methylation beta values weighted by their estimated effect sizes in the training dataset (i.e., AIBL dataset). In our best performing model involving MRS based on prioritized CpGs from cross-tissue analysis, the results showed when tested individually, the models that included age and sex, cell types, or MRS alone had AUCs of 0.649, 0.576, and 0.614, respectively. When combined with cell types or MRS, the model with age and sex improved prediction accuracy to 0.663 and 0.688, respectively (see Figure 4 below). The best performing model included age, sex, MRS, and cell types (AUC = 0.696, 95% CI: 0.616 - 0.770), significantly more predictive than a random classifier with an AUC of 0.5 (P -value = 2.78×10^{-5}).

We added a new section and two new figures in the manuscript on out-of-sample validation of AD-associated DNAm differences in an external cohort to discuss these results. Please see details in (1)-(4) under "Revisions" below.

Revisions

(1) In Results, added a new section on out-of-sample validation

Out-of-sample validations of AD-associated DNA methylation differences in an external cohort

To evaluate the feasibility of the identified methylation differences for predicting AD diagnosis, we next performed out-of-sample validations using an external DNA methylation dataset generated by the AddNeuroMed study, which included 83 AD cases and 88 control samples with ages greater than 65 years⁵. To this end, we computed the methylation risk scores (MRS)⁴, which were shown to have excellent discrimination of smoking status, and moderate discrimination of obesity, alcohol consumption, HDL cholesterol⁶, and amyotrophic lateral sclerosis case-control status⁷ recently.

More specifically, for each sample in the AddNeuroMed dataset, we computed MRS by summing the methylation beta values of the prioritized CpGs weighted by their estimated effect sizes in the AIBL dataset. Several logistic regression models were estimated using the AIBL dataset and then tested on the AddNeuroMed dataset. We considered logistic regression models with three sources of variations that might affect prediction for AD diagnosis: known clinical factors (i.e., age and sex), estimated cell-type proportions for each sample, and MRS. When tested individually, the models that included age and sex, cell types, or MRS alone had AUCs of 0.649, 0.576, and 0.614, respectively (Figure 3). When combined with estimated cell types or MRS, prediction performance for the model with clinical factors (i.e., age and

sex) improved to AUCs of 0.663 and 0.688, respectively. Notably, MRS was computed based on fewer than one hundred CpGs, while the six variables corresponding to cell type proportions were estimated using all CpGs on the array. The best performing model included age, sex, MRS, and cell types (AUC = 0.696, 95% CI: 0.616 - 0.770) (Figures 3-4), significantly more predictive than a random classifier with an AUC of 0.5 (P -value = 2.78×10^{-5}).

The same logistic regression model trained using both ADNI and AIBL datasets (instead of AIBL alone) performed slightly worse with an AUC of 0.678, which might be due to batch effect due to different training datasets. The model that included MRS based on AD-associated CpGs (instead of prioritized CpGs) also performed worse with an AUC of 0.609, probably because CpGs with cross-tissue differences also leveraged information from additional brain samples datasets. In addition to MRS, we also evaluated the performance of MPS (methylation PC scores) described above, which sums methylation beta values in the testing dataset weighed by loadings of the first principal component in PCA analysis. The best performing logistic regression model involving MPS was also estimated using the AIBL dataset, included variables age, sex, cell types, and MPS computed based on the same 91 prioritized cross-tissue CpGs, and achieved an AUC of 0.662.

(2) Added Figure 3 and Figure 4 for results of out-of-sample validation

Figure 3 Performance of different logistic regression models for predicting AD diagnosis in out-of-sample validation. The training samples included 135 AD cases and 356 control samples from the AIBL dataset and the testing samples included 83 AD cases and 88 control samples from the AddNeuromed dataset. MRS was computed as the sum of methylation beta values for 91 prioritized CpGs from cross-tissue analysis weighted by their estimated effect sizes in the AIBL dataset. Abbreviation: AUC = Area Under ROC curve

Figure 4 Receiver Operating Characteristic curves (ROCs) for logistic regression models predicting AD diagnosis in out-of-sample validation using the AddNeuroMed dataset (83 AD cases, 88 controls). The training dataset included 135 AD cases and 356 control samples from the AIBL dataset. The best performing logistic regression model (AUC = 0.696) included methylation risk score (MRS), age, sex, and estimated cell-type proportions, where MRS was computed as the sum of methylation beta values for prioritized CpGs weighted by their estimated effect sizes in the AIBL dataset.

(3) In Discussion, 2nd to the last paragraph, added text to discuss out-of-sample validation

This study has several limitations.... Fifth, the MRS-based risk prediction model could also be further improved. Because DNA methylation samples in the testing dataset (AddNeuroMed) were measured by 450k arrays, which are different from the EPIC arrays used by the AIBL study, we only included CpGs that mapped to both arrays in the computation of our MRS. The performance of our MRS-based prediction models can be assessed more accurately using future testing dataset measured by the EPIC arrays. Also, in

the out-of-sample validation analysis, we did not include other important factors such as APOE genotype, which might also significantly predict AD diagnosis⁸ because we did not have access to APOE information in the AIBL and AddNeuroMed datasets. Our internal validation using the ADNI dataset suggested additionally including APOE (ϵ 4 allele) into our best performing logistic regression model, which included MRS, age, sex, and estimated cell types (Figures 3-4), might substantially improve prediction performance. More specifically, a 10-fold cross-validation using the ADNI dataset showed the estimated average AUCs for the best performing logistic regression models with and without APOE status were 0.691 and 0.810, respectively (Supplementary Table 16). Finally, the associations we identified do not necessarily reflect causal relationships. Additional studies are needed to establish the causality of the nominated DNA methylation markers.

(4) In Methods, last paragraph

Out-of-sample validation

The best-performing risk prediction model was trained using samples from the AIBL dataset and tested on the AddNeuroMed dataset. More specifically, first, we computed the Methylation Risk Scores (MRS) as the sum of methylation beta values (for prioritized CpGs in Supplementary Table 4) weighted by their estimated effect sizes (i.e., parameter estimate for methylation beta values in logistic regression after bacon-correction) in the AIBL dataset⁴. We included 91 prioritized CpGs that were available in the AddNeuroMed dataset from GEO. Next, the logistic regression model $\text{logit}(\text{Pr}(\text{AD})) \sim \text{MRS} + \text{age} + \text{sex} + \text{B} + \text{NK} + \text{CD4T} + \text{CD8T} + \text{Mono} + \text{Neutro}$ was fitted to the AIBL dataset using `glm()` function, and `predict.glm()` was used to apply the logistic regression model to AddNeuroMed dataset. The last six variables in the logistic regression model correspond to estimated proportions of different blood cell types of B-cells, Natural Killer (NK) cells, CD4+ T-cells, CD8+ T-cells, Monocytes, and Neutrophils, obtained using the EpiDish R package⁹. The R package pROC was used to estimate receiver operating characteristic curves (ROCs) and area under the ROC curves (AUCs). Similarly, logistic regression models with a subset of the variables in the above model (e.g., only age and sex) were similarly developed using the AIBL dataset and tested on the AddNeuroMed dataset. To determine if a logistic regression model predicted AD diagnosis significantly better than chance, we used the Wilcoxon rank-sum test to compare estimated probabilities for AD cases versus controls¹⁰. To assess the added prediction accuracy of APOE gene, we performed internal validations (i.e., 10-fold cross-validations) that compared our best performing model $\text{logit}(\text{Pr}(\text{AD})) \sim \text{MRS} + \text{age} + \text{sex} + \text{B} + \text{NK} + \text{CD4T} + \text{CD8T} + \text{Mono} + \text{Neutro}$ with the model that additionally included APOE (ϵ 4 allele genotype) using the ADNI dataset. To obtain an independent set of samples, only the last visit of each subject in the ADNI dataset was used for this analysis. The function `createFolds()` in `caret` R package was

used to divide the data into 10 folds. Average AUCs over the 10 iterations in the 10-fold cross-validations for the models with and without APOE were then estimated and compared.

References

1. Garcia-Esparcia, P. *et al.* Altered mechanisms of protein synthesis in frontal cortex in Alzheimer disease and a mouse model. *Am J Neurodegener Dis* **6**, 15-25 (2017).
2. Hernandez-Ortega, K., Garcia-Esparcia, P., Gil, L., Lucas, J.J. & Ferrer, I. Altered Machinery of Protein Synthesis in Alzheimer's: From the Nucleolus to the Ribosome. *Brain Pathol* **26**, 593-605 (2016).
3. Tammineni, P., Ye, X., Feng, T., Aikal, D. & Cai, Q. Impaired retrograde transport of axonal autophagosomes contributes to autophagic stress in Alzheimer's disease neurons. *Elife* **6**(2017).
4. Huls, A. & Czamara, D. Methodological challenges in constructing DNA methylation risk scores. *Epigenetics* **15**, 1-11 (2020).
5. Roubroeks, J.A.Y. *et al.* An epigenome-wide association study of Alzheimer's disease blood highlights robust DNA hypermethylation in the HOXB6 gene. *Neurobiol Aging* **95**, 26-45 (2020).
6. McCartney, D.L. *et al.* Epigenetic prediction of complex traits and death. *Genome Biol* **19**, 136 (2018).
7. Nabais, M.F. *et al.* Significant out-of-sample classification from methylation profile scoring for amyotrophic lateral sclerosis. *NPJ Genom Med* **5**, 10 (2020).
8. Sims, R., Hill, M. & Williams, J. The multiplex model of the genetics of Alzheimer's disease. *Nat Neurosci* **23**, 311-322 (2020).
9. Zheng, S.C., Breeze, C.E., Beck, S. & Teschendorff, A.E. Identification of differentially methylated cell types in epigenome-wide association studies. *Nat Methods* **15**, 1059-1066 (2018).
10. Mason, S.J. & Graham, N.E. Areas beneath the relative operating characteristics (ROC) and relative operating levels (ROL) curves: Statistical significance and interpretation. *Quarterly Journal of the Royal Meteorological Society* **128**, 2145-2166 (2002).

REVIEWERS' COMMENTS

Reviewer #1 (Remarks to the Author):

I thank the authors for their truly thorough response. I was impressed by the level of detail and genuine consideration of reviewer comments. I feel that the manuscript has greatly improved. I have one final comment:

The authors write that 'To obtain an independent set of samples, only the last visit of each subject in the ADNI dataset was used for this analysis'. This statement is a little confusing, as I assumed that the ADNI testing data was completely independent of the training data. Was this done because APOE was only available in the same data used for training? If so, then this could be further clarified in a figure (training/testing sample and internal validation including APOE genotype) or simply in text. Maybe it would be enough to remove to section on APOE-internal validation from the section on 'out-of-sample validation' to a separate section on 'internal validation including APOE' (or similar).

(reviewed by Esther Walton)

Reviewer #3 (Remarks to the Author):

I had previously lightly suggested the authors attempt a biomarker prediction on an independent sample if it was no out of scope and am pleased they attempted to preform one. I'm guessing because it is not in the paper that RIN3 alone didn't add any predictive ability. It is understandable why the authors chose to use an MRS approach using Meta-analysis weights, although my personal feeling is that a weighted sum approach akin to GWAS may lose the potential for co-regulated genes and/or interactions within the 'important' gene list to contribute to model prediction. I am left wondering if including those 91 overlapping beta values available in the 450K (or the top 8) into a machine learning model (random forest or support vector regression) would perform similarly. Still, what the authors chose is a valid approach and I see no problems with publishing it. Notably, the MRS only added a tiny amount of additional sensitivity and specificity as compared to just looking at blood cell types, age, and sex, for example (c-stat increase of $\sim .39$).

Other than that, the paper remains a great addition to the field.

REVIEWERS' COMMENTS

Reviewer #1 (Remarks to the Author):

I thank the authors for their truly thorough response. I was impressed by the level of detail and genuine consideration of reviewer comments. I feel that the manuscript has greatly improved. I have one final comment:

The authors write that 'To obtain an independent set of samples, only the last visit of each subject in the ADNI dataset was used for this analysis'. This statement is a little confusing, as I assumed that the ADNI testing data was completely independent of the training data. Was this done because APOE was only available in the same data used for training? If so, then this could be further clarified in a figure (training/testing sample and internal validation including APOE genotype) or simply in text. Maybe it would be enough to remove to section on APOE-internal validation from the section on 'out-of-sample validation' to a separate section on 'internal validation including APOE' (or similar).

(reviewed by Esther Walton)

We appreciate this reviewer's helpful comments, which helped us to significantly improve the manuscript. Thank you very much! To clarify, among the three public datasets (AIBL, AddNeuroMed, ADNI) we analyzed, APOE information was only available in the ADNI dataset. As this reviewer suggested, we added text to clarify that APOE is only available in the ADNI dataset, removed the APOE-internal validation from the "out-of-sample validation" section, and added a separate section on "Internal validation to assess the impact of APOE genotype".

Revision – in Methods, last paragraph

Internal validation to assess the impact of APOE genotype

Among the three public datasets (AIBL, AddNeuroMed, ADNI) we analyzed, the APOE genotype was only available for the ADNI dataset. To assess the added prediction accuracy of APOE gene, we performed internal validations (i.e., 10-fold cross-validations) using the ADNI dataset, by comparing our best performing model $\text{logit}(\text{Pr}(AD)) \sim \text{MRS} + \text{age} + \text{sex} + \text{B} + \text{NK} + \text{CD4T} + \text{CD8T} + \text{Mono} + \text{Neutro}$ with the model that additionally included APOE ($\epsilon 4$ allele genotype). To obtain an independent set of samples, only the last visit of each subject in the ADNI dataset was used for this analysis. The function `createFolds()` in the `caret` R package was used to divide the data into ten folds. Average AUCs over the ten iterations in the 10-fold cross-validations for the models with and without APOE were then estimated and compared.

Reviewer #3 (Remarks to the Author):

I had previously lightly suggested the authors attempt a biomarker prediction on an independent sample if it was no out of scope and am pleased they attempted to preform one. I'm guessing because it is not in the paper that RIN3 alone didn't add any predictive ability. It is understandable why the authors chose to use an MRS approach using Meta-analysis weights, although my personal feeling is that a weighted sum approach akin to GWAS may lose the potential for co-regulated genes and/or interactions within the 'important' gene list to contribute to model prediction. I am left wondering if including those 91 overlapping beta values available in the 450K (or the top 8) into a machine learning model (random forest or support vector regression) would perform similarly. Still, what the authors chose is a valid approach and I see no problems with publishing it. Notably, the MRS only added a tiny amount of additional sensitivity and specificity as compared to just looking at blood cell types, age, and sex, for example (c-stat increase of $\sim .39$).

Other than that, the paper remains a great addition to the field

We thank this reviewer again for the enthusiasm for our study! The reviewer raised an important point on using machine learning approaches to further improve the performance of the methylation-based prediction models. As this reviewer suggested, we implemented random forest and support vector regression, with the 91 CpGs prioritized by cross-tissue analysis, age, sex, and estimated cell-type proportions as predictors, using AIBL as training dataset, and AddNeuroMed as testing dataset. The analysis was performed using R package tidymodels, with default parameters for RF and SVM. Our results showed RF and SVM achieved AUCs of 0.61 and 0.64, respectively (Figure 1 below). Using cross-validation in the training dataset to select model parameters did not further improve prediction performance. Therefore, we still kept the presentation of our final results in the manuscript the same as before using logistic regression models, given their better interpretability and better performance with an AUC of 0.696 (Figure 4 in manuscript).

Figure 1 Receiver Operating Characteristic curves (ROCs) for random forests (RF) and support vector regression models (SVM) predicting AD diagnosis in out-of-sample validation using the AddNeuroMed (83 AD cases, 88 controls). The training samples included 135 AD cases and 356 control samples from the AIBL dataset. Both RF and SVM models included the 91 CpGs prioritized by cross-tissue analysis, age, sex, and estimated cell-type proportions as predictors. The RF and SVM models achieved AUCs of 0.61 and 0.64, respectively.